# Genetic architecture of telomere length in 462,666 UK Biobank whole-genome sequences

Oliver S. Burren[1,31], Ryan S. Dhindsa [2,31], Sri V. V. Deevi [1,31], Sean Wen[1], Abhishek Nag[1], Jonathan Mitchell[1], Fengyuan Hu[1], Douglas P. Loesch[1], Katherine R. Smith [1], Neetu Razdan[3], Henric Olsson [4], Adam Platt [5], Dimitrios Vitsios [1], Qiang Wu[2,6], AstraZeneca Genomics Initiative*, Veryan Codd[7], Christopher P. Nelson[7], Nilesh J. Samani[7], Ruth E. March[8], Sebastian Wasilewski[1], Keren Carss [1], Margarete Fabre[1,9,10], Quanli Wang[2], Menelas N. Pangalos[11] & Slavé Petrovski [1,12] ✉

Telomeres protect chromosome ends from damage and their length is linked with human disease and aging. We developed a joint telomere length metric, combining quantitative PCR and whole-genome sequencing measurements from 462,666 UK Biobank participants. This metric increased SNP heritability, suggesting that it better captures genetic regulation of telomere length. Exome-wide rare-variant and gene-level collapsing association studies identified 64 variants and 30 genes significantly associated with telomere length, including allelic series in *ACD* and *RTEL1*. Notably, 16% of these genes are known drivers of clonal hematopoiesis—an age-related somatic mosaicism associated with myeloid cancers and several nonmalignant diseases. Somatic variant analyses revealed gene-specific associations with telomere length, including lengthened telomeres in individuals with large *SRSF2*-mutant clones, compared with shortened telomeres in individuals with clonal expansions driven by other genes. Collectively, our findings demonstrate the impact of rare variants on telomere length, with larger effects observed among genes also associated with clonal hematopoiesis.

Telomeres are repetitive nucleotide sequences that protect the ends of chromosomes from degradation and are thus crucial for maintaining genomic integrity. In somatically dividing cells, telomeres shorten with each replication cycle until they reach a critical length that triggers cellular senescence and ultimately cell death[1,2]. Telomere length demonstrates considerable interindividual variability modulated by heritable[3,4], environmental and lifestyle factors such as smoking behavior and stress[5]. Rare germline mutations linked to telomere shortening have been associated with severe diseases, including premature aging syndromes, interstitial lung disease and immunodeficiencies[1,6,7].

More subtle reductions in telomere length have been associated with common, age-related diseases, such as coronary artery disease[8]. Although telomere length is heritable, our current understanding of its genetic determinants has been largely limited to the study of common variants. A greater understanding of the genetic determinants of telomere length would provide insights into disease pathogenesis, thereby identifying potential new therapeutic targets.

High-throughput telomere length assays have been developed to understand telomere biology at the population level. One such method uses quantitative PCR (qPCR) to measure the relative abundance of

A full list of affiliations appears at the end of the paper. *A list of authors and their affiliations appears at the end of the paper.
✉e-mail: slav.petrovski@astrazeneca.com

telomere sequences compared with a reference sequence[9]. More recently introduced in silico methods, such as TelSeq, measure average telomere length from whole-genome sequencing (WGS) data[10]. The advances in genome sequencing of population-scale biobanks provides unprecedented opportunities to leverage these approaches to study the genetic architecture of telomere length and ultimately its impact on human health at a population scale. In a recent study of over 472,174 UK Biobank (UKB) participants, a microarray-based, genome-wide association study (GWAS) identified >100 independent common variant loci associated with qPCR telomere length measurements[8]. By combining these measurements with whole-exome sequencing (WES) data across 418,401 individuals, Kessler et al. identified rare-variant associations for several previously established genes[11]. Another study applied the TelSeq algorithm to estimate telomere length from the whole-genome sequences of 109,122 multiancestry individuals from the TopMed program and identified 36 associated loci, which largely overlap those identified by qPCR-based measures[12].

In the present study, we leverage a larger sample size of WGS data from 490,397 multiancestry UKB participants to study the genetic architecture of telomere length, including contributions from both rare and common variants. Moreover, in comparing qPCR- and WGS-derived telomere length estimates in the same individuals, we observed that combining both measurements into a single statistical metric significantly improved the accuracy of telomere length estimates and empowered discovery potential.

## Results

### A combined telomere length metric increases heritability

Of the 490,397 UKB participants with WGS data, we took forward for analysis 462,666 UKB samples (94%) that met our quality control (QC) thresholds (Methods) and for whom qPCR telomere length estimates were also available (Supplementary Table 1 and Extended Data Fig. 1). As an alternative method for estimating telomere length, we also used TelSeq (Supplementary Fig. 1), which estimates telomere length from the WGS data[10].

As expected, telomere length estimated from TelSeq and qPCR were both significantly associated with age, sex and ancestry (Supplementary Table 2 and Extended Data Fig. 2). It is interesting that the qPCR- and coverage-adjusted TelSeq telomere length estimates were only moderately correlated ($r^2 = 0.29$; Fig. 1a) after consideration of potential sequencing confounders (Extended Data Fig. 3, Supplementary Figs. 2–5, Supplementary Table 3 and Supplementary Notes 1 and 2). In a joint model, the association between each of the metrics and age remained highly significant, suggesting that each captures additional information. We derived a principal component analysis (PCA) linear combination[13] incorporating both qPCR and adjusted TelSeq (Fig. 1b and Extended Data Fig. 4). Use of the first principal component, PC1, demonstrated a significant ($P < 1 \times 10^{-16}$, linear regression, two-sided unadjusted) performance gain in predicting age compared with models employing either of the individual measures (Supplementary Fig. 6).

We first sought to determine common variants (minor allele frequency (MAF) > 0.1%) associated with telomere length, focusing on 438,351 non-Finnish European (NFE) broad genetic ancestry individuals with array-based imputed genotypes available (Supplementary Table 1 and Extended Data Fig. 1). Using REGENIE[14], we performed a common variant GWAS of telomere length estimates derived from qPCR, WGS, PC1 or PC2 (Fig. 1c and Methods) replicating all signals from Codd et al.[8] (Supplementary Note 3). Linkage disequilibrium (LD)-score regression[15] revealed that the PC1 vector had the highest heritability ($h^2 = 0.099$, s.e.m. ± 0.010; Supplementary Table 4), suggesting that the combined telomere length metric explains more telomere length variance resulting from genetic variation than either qPCR or TelSeq alone.

We undertook single-variant fine-mapping for all significant ($P < 5 \times 10^{-8}$) loci (excluding the major histocompatibility region) in the qPCR, TelSeq and PC1 GWAS. The PC1 telomere length score resulted in

smaller 95% credible SNP sets (median = 8) compared with the separate qPCR and WGS GWASs (median = 12 and 11, respectively), highlighting that PC1 can more effectively identify potentially causal variants. In total for PC1, we identified 192 significant ($P < 5 \times 10^{-8}$) loci (Supplementary Tables 5 and 6), 70 of which were not within 1 Mb of a previously implicated locus. Associations at known loci were also stronger with PC1 compared with qPCR or TelSeq, further demonstrating the value of the combined metric (Extended Data Fig. 5).

There were also 22 significant loci identified in the PC2 GWAS (Supplementary Tables 5 and 6), most of which were driven exclusively by a single telomere length metric (Supplementary Fig. 7). Moreover, 50% of these associations ($n = 11$ of 22; 3q29:*LMLN*, 5p15.33:*PLEKHG4B*, 6p25.3:*DUSP22*, 7q36.3:*VIPR2*, 8p23.3:*ZNF596*, 11p15.5:*BET1L*, 16q24.3:*PRDM7*, 17p13.13:*DOC2B*, 18q23:*PARD6G*, 20p13:*DEFB125* and 20q13.33:*RTEL1*) were peritelomeric (<2 Mb). There was one qPCR association at 11p15.4 (rs1609812) proximal to *HBB* ($P = 6.8 \times 10^{-60}$, $\beta = -0.05$ (confidence interval (CI), −0.05 to −0.04)), which is used as the reference gene to normalize the qPCR assay and has been previously thought to be driven by artefactual technical signals[8]. Consistent with this being a putative artifact, this locus was not significant in the TelSeq GWAS ($P = 0.99$, $\beta = 0$ (−0.005 to 0.005); Supplementary Fig. 8). Collectively, these results demonstrate the superior performance of a linear combination of telomere length metrics to detect associations and further highlight PC2's potential to flag spurious associations.

### Rare-variant-level associations with telomere length

We observed that rare variants have demonstrably larger effects on telomere length than common variants and have also been implicated in numerous telomere-related diseases. In the present study, we focused on protein-coding variants observed in WGS data from 439,351 UKB participants of NFE broad genetic ancestry to examine the effect of rare variation on PC-derived telomere length estimates. After removing individuals with known hematological malignancies at sampling ($N = 3,073$), we performed both variant-level (exome-wide association study (ExWAS)) and gene-level (rare-variant-aggregated collapsing) analyses[16]. We observed high concordance ($r^2 = 0.99$) between the effect sizes for the common variants included in the ExWAS and our separate common variant GWAS (microarray genotyping) analyses. Genomic inflation was also well controlled with a median $\lambda_{GC} = 1.07$ (Supplementary Fig. 9).

We restricted our downstream analyses of the ExWAS to rare (MAF < 0.1%) exonic variants that were too rare to be well represented in the GWAS. Based on our previously identified significance threshold of $P \leq 1 \times 10^{-8}$ (ref. [16]), there were 62 significant rare-variant germline associations across 19 distinct genes (Fig. 2a and Supplementary Table 7) for PC1 after excluding variants that were also significantly associated with PC2 (Supplementary Fig. 10). Although all of the variants except 8-84862338-A-G (*RALYL*.p.Ala165Ala, $P = 4.8 \times 10^{-11}$, $\beta = 2.24$ (1.57–2.90)) overlapped with a previously identified GWAS locus, the absolute effect sizes observed for the ExWAS analyses were generally significantly greater than that previously reported for the same loci. Of the 62 rare-variant germline signals, 16% (10 of 62) were only significantly associated with PC1 and not underlying qPCR or TelSeq measurements.

Thirty-nine germline rare variants were associated with longer telomere length and clustered in components of the CST (*CTC1*) and Shelterin (*ACD*, *TERF1* and *TINF2 POT1*) complexes, both of which function to protect telomere ends and regulate interactions with telomerase. Of these, ten were protein-truncating variants (PTVs) in *CTC1*, *POT1*, *SAMHD1*, *TINF2* and *TERF1*, all of which are genes implicated in telomere-associated diseases. It is interesting that the two PTVs in *CTC1* (17-8237439-GCTTT-G p.Lys242fs: $P = 1.35 \times 10^{-24}$, $\beta = 0.54$ (0.44–0.65); and 17-8229438-AG-A p.Leu1007fs: $P = 4.12 \times 10^{-11}$, $\beta = 0.53$ (0.37–0.69)) have both been implicated in compound, heterozygous, recessive, cerebroretinal microangiopathy with calcifications and cysts (CMCC, also known as Coats plus syndrome), which is associated with shorter

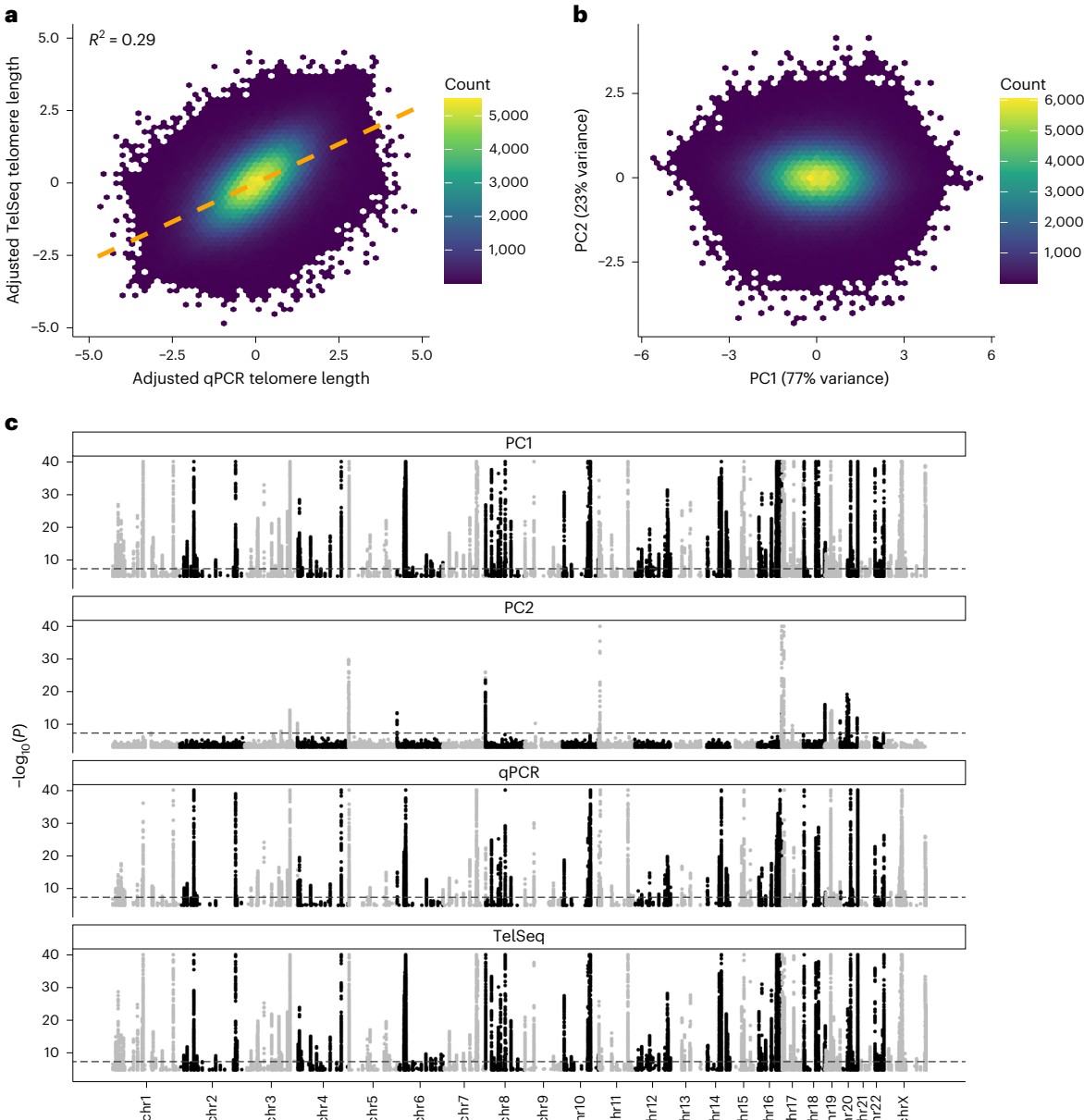

**Fig. 1 | Combining telomere length metrics improves genetic discovery.**
**a**, Correlation between inverse normal transformed qPCR and WGS TelSeq telomere length metrics. The orange dashed line indicates a linear model line of best fit. **b**, Biplot for PCA of qPCR and TelSeq telomere length metrics. **c**, Manhattan plot of common variant analysis of PC1, PC2, qPCR and TelSeq in the NFE broad genetic ancestry group. *P* values (two-sided, unadjusted) are derived from REGENIE analysis of 438,351 independent samples; the dotted line indicates $P = 5 \times 10^{-8}$ and for clarity *y* axes are truncated at $P < 1 \times 10^{-40}$.

telomeres[17,18]. Our results indicate that, outside the context of nullizygosity, this PTV is associated with longer telomere length, concordant with prior observations of CTC1 depletion promoting excessive telomerase activity[19]. We also observed four PTVs associated with telomere length in *POT1*, which is associated with familial glioma, familial melanoma, cardiac angiosarcoma and chronic lymphocytic leukemia[20–24].

Remarkably, the remaining 23 rare nonsynonymous germline variants associated with shorter telomere length and were clustered in genes previously associated with autosomal dominant dyskeratosis congenita and/or pulmonary fibrosis (IPF) (*ACD* (Online Mendelian Inheritance in Man (OMIM): 609377), *PARN* (OMIM: 604212), *RTEL1* (OMIM: 608833), *NAF1* (OMIM: 620365) and *TERT* (OMIM: 613989)). In both *ACD and RTEL1*, we observed independent, rare, nonsynonymous variants with opposing effects, indicating a possible allelic series in these two genes. For example, in *ACD* three rare missense

variants clustering within the *POT1*-binding domain (16-67659017-C-T p.Val269Met, 16-67659046-C-A p.Arg259Leu and 16-67659234-T-C p.Asn246Ser) were associated with increased telomere length and one (16-67660036-C-T p.Asp120Asn) in the amino-terminal oligonucleotide/oligosaccharide-binding (OB) domain that acted in the opposite direction (Table 1). *ACD* encodes TPP1, a key component of the six-protein Shelterin complex. Consistent with our results and previous studies[25–28], a recent mutagenesis revealed that mutations that disrupt *POT1* binding promote ectopic initiation of ATR (ataxia–telangiectasia-mutated (ATM) and Rad3-related)- and ATM-mediated DNA damage-repair programs, resulting in longer telomeres[29]. Reciprocally, mutations within the N-terminal OB are associated with disrupted telomerase recruitment, leading to progressively shorter telomere length[29], mirroring the effect of the 16-67660036-C-T variant that we detected in this region.

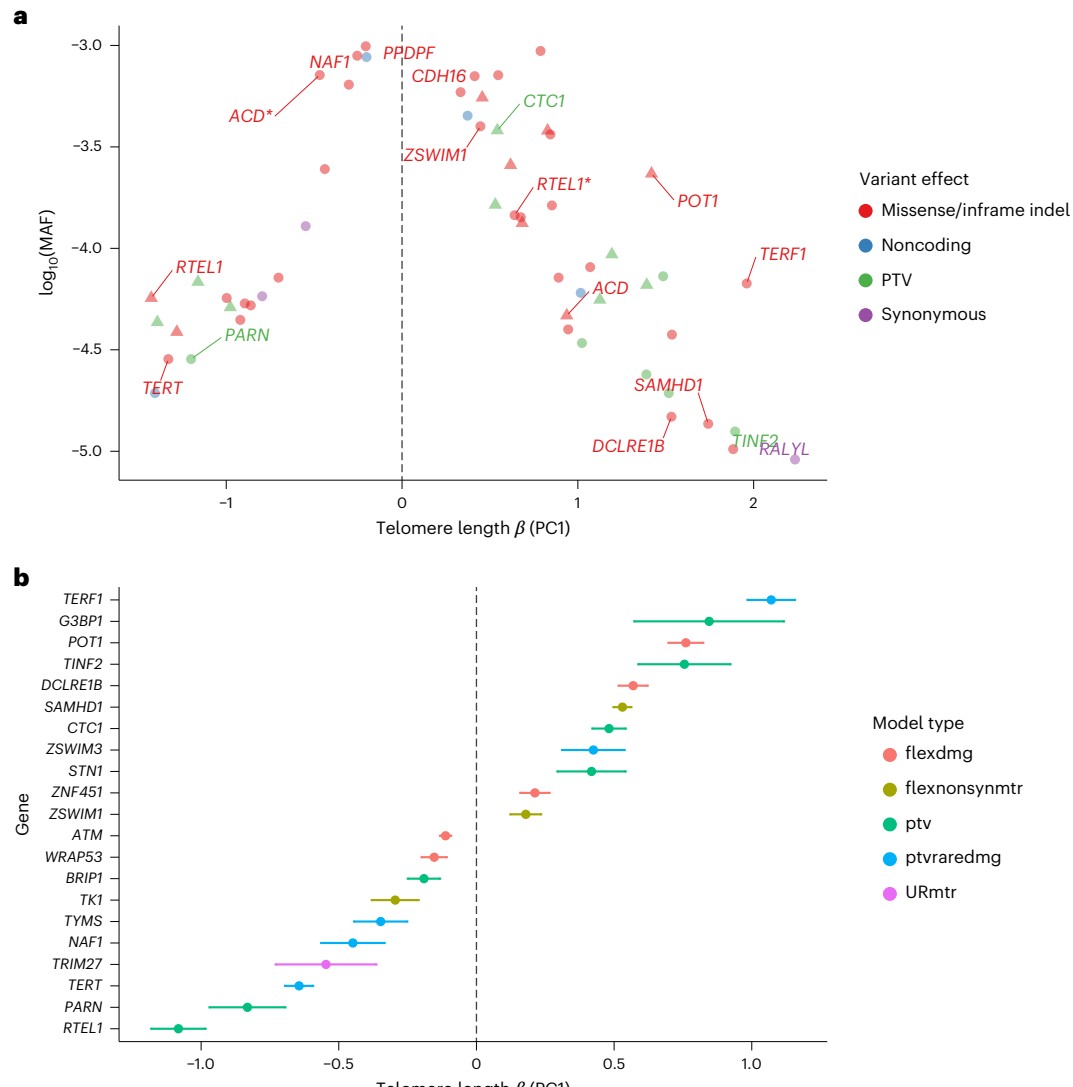

**Fig. 2 | Rare-variant analysis of telomere length. a**, ExWAS analysis of PC1 telomere length in the NFE broad genetic ancestry group, showing only rare germline variants that are significant ($P \le 1 \times 10^{-8}$) for PC1 and not PC2. For clarity the variant with the largest effect for a gene is labeled, variants with opposing effect size in the same gene are starred and triangles indicate *HGMD* pathogenic variants. *P* values (two-sided, unadjusted) were calculated from fitting a linear regression model. Color represents the functional effect of the variant on protein. **b**, Collapsing analysis of PC1, showing the most significant ($P \le 1 \times 10^{-8}$) association for a gene over all qualifying variant models (Supplementary Table 9). Associations driven by putative somatic variants are excluded. Colors represent the qualifying variant model used in collapsing analysis. Error bars represent 95% CIs. For both plots $N = 436,410$ independent samples.

Although less frequent than common variants, rare variants can still be correlated as a result of LD and, to resolve signal independence, we performed conditional analyses (Methods) and found that four of our signals in *SOGA1*, *PCIF1*, *MYH11* and *MTSS1L* are probably the result of LD contamination. For example, the variant in *SOGA1* (20-36810011-C-T p.Ala852Thr: $P = 1.9 \times 10^{-32}$, $\beta = 0.46$ (0.38–0.54)) is probably due to LD with an *SAMHD1* 20-36898455-C-G signal (Supplementary Table 8).

**Rare-variant gene-level collapsing analysis**

We performed gene-level collapsing analyses to identify genes associated with telomere length through the aggregated presence of variants too rare and thus underpowered to be individually discovered in ExWAS analyses. We employed ten qualifying variant (QV) models[16] (Supplementary Table 9), and association statistics were well calibrated with a median $\lambda_{GC} = 1.12$ (Supplementary Fig. 11). After filtering putative somatic signals, we identified 20 genes significantly ($P \le 1 \times 10^{-8}$) associated with PC1 telomere length, 2 (10%) of which were uniquely

identified in PC1 and not the individual qPCR or TelSeq statistics (Fig. 2b, Supplementary Table 10 and Extended Data Fig. 6).

Sixteen of the gene-level signals arose from the rare protein-truncating 'PTV' QV model. Six of these genes were associated with telomere shortening (*ATM*, *BRIP1*, *NAF1*, *PARN*, *RTEL1* and *TERT*), five of which have been implicated in known telomere-related clinical diseases, including IPF[30–32], Fanconi's anemia[33] and dyskeratosis congenita[34]. The remaining ten PTV collapsing model signals were associated with longer telomere length. Seven of these ten genes have established biological roles in protection from telomere length attrition (*POT1*, *TERF1*, *TFIN2*, *CTC1* and *STN1*), DNA repair (*DCLRE1B*; formerly *APOLLO*) and thymidine nucleotide metabolism (*SAMHD1* (ref. 35)).

Three genes significantly associated with telomere lengths in the rare PTV collapsing model have not been previously described in the context of telomere length biology. Of the two associated with longer telomere length, *G3BP1* ($P = 1.2 \times 10^{-9}$, $\beta = 0.85$ (0.57–1.12)), encodes an RNA-binding protein involved in RNA metabolism regulation and stress

**Table 1 | Rare variants in *ACD* modulating telomere length**

| RS no. | Variant ID | MAF | Effect (95% CI) | P value | Consequence[a] | Domain |
|---|---|---|---|---|---|---|
| rs139438549[b] | 16-67658960-T-C | 0.001 | 0.43 | $9.9×10^{-11}$ | Thr205Ala | |
| rs145007645 | 16-67659046-C-A | $1.6×10^{-4}$ | 0.85 (0.69–1.01) | $6.4×10^{-26}$ | Arg176Leu | |
| rs370512338 | 16-67659234-T-C | $3.8×10^{-4}$ | 0.83 (0.72–0.93) | $1.2×10^{-55}$ | Asn163Ser | *POT1*-binding domain |
| rs249052024 | 16-67659240-G-A | $6.4×10^{-4}$ | −0.30 (−0.38 to −0.22) | $9.5×10^{-14}$ | Ser161Leu | |
| rs142662151 | 16-67660036-C-T | $7.1×10^{-4}$ | −0.47 (−0.54 to −0.39) | $6.4×10^{-34}$ | Asp37Asn | OB1 |

Test statistics (two-sided, unadjusted) are derived from a linear model using the PC1 telomere length metric across 436,410 independent participants of broad NFE genetic ancestry. Effect and 95% CIs are on the unit scale. [a]Protein coordinates with respect to UniProt (Q96AP0) canonical transcript ENST00000620761.6. [b]Also detected through our GWAS.

granule formation[36]. It is also known to bind guanine quadruplexes, which are a substrate for human telomerase[37,38]. The other gene, *ZNF451* ($P = 1.2 × 10^{-11}$, $\beta = 0.36$ (0.25–0.46)), encodes a zinc finger protein that acts as a SUMO (small ubiquitin-like modifier) ligase and a DNA repair factor that controls cellular responses to TOP2 damage[39]. Finally, PTVs in *BRIP1* ($P = 7.5 × 10^{-8}$, $\beta = -0.18$ (−0.24 to −0.12)) were associated with shorter telomere length. *BRIP1* is a DNA helicase involved in homologous recombination and has been associated with ovarian cancer, breast cancer and Fanconi's anemia[40–42].

There were several other, previously unreported, significant associations that arose in the QV models that included PTV effects alongside putatively damaging missense variants. *TK1* (flexnonsynmtr (flexdmg with additional MTR (missense intolerant regions) filter), $P = 1.08 × 10^{-11}$, $\beta = -0.30$ (−0.38 to −0.21)) and *TYMS* (flexdmg (flexible nonsynonymous), $P = 1.70 × 10^{-12}$, $\beta = -0.35$ (−0.44 to −0.25)), which have also been observed as hits in a clustered regularly interspaced short palindromic repeats (CRISPR)–Cas9 screen for telomere length[35] and causally associated with dyskeratosis congenita[43], were associated with reduced telomere length. *WRAP53* (flexdmg, $P = 5.9 × 10^{-9}$, $\beta = -0.14$ (−0.19 to −0.09)), which encodes a component of the telomerase holoenzyme complex, was also associated with decreased telomere length. The *ZSWIM1* (flexnonsynmtr, $P = 9.5 × 10^{-9}$, $\beta = 0.17$ (0.11–0.22)) and *ZSWIM3* (ptvraredmg (union of PTV and rare damaging variants), $P = 2.41 × 10^{-13}$, $\beta = 0.43$ (0.31–0.53)) zinc finger proteins were associated with increased telomere length. *ZSWIM1* (flexnonsynmtr, $P = 9.45 × 10^{-9}$, $\beta = 0.17$ (0.11–0.22)), which was also an ExWAS hit, and *ZSWIM3* are in proximity with each other, sitting within a peritelomeric GWAS locus. We thus performed a leave-one-out (LOO) analysis (Methods), which showed that no individual variants in *ZWIM1* and/or *ZSWIM3* were responsible for driving either gene-level association (Supplementary Fig. 12). Moreover, conditional analysis indicated that both *ZSWIM1* and *ZSWIM3* associations were independent of each other and of the 20-45884012-G-A *ZSWIM1* missense variant identified from our ExWAS analysis. Altogether, the rare-variant, aggregated, gene-level collapsing analysis framework uncovered several loci that were not detectable in the variant-level analyses.

### Causal associations between the proteome and telomere length

We integrated protein quantitative trait locus (pQTL) data from the UKB Pharma Proteomics Project that examined genetic associations across approximately 3,000 plasma proteins[44] with our telomere length PC1 genetic associations. Across all PC1 GWAS significant loci, we identified 2,905 overlapping pQTLs ($P < 1.7 × 10^{-11}$) (Supplementary Table 11). We used coloc[45] to assess each of these and found strong evidence for a shared causal variant modulating both telomere length and plasma protein abundance at 266 (9%) of these overlaps. Of these, 10 were colocalizations in *cis* and 256 were in *trans*. For the *cis* signals we used pQTLs as instruments in a Mendelian randomization (MR) analysis

(Methods) to assess whether plasma proteome abundance might be causally related to telomere length. We found evidence for a causal interrelationship across nine protein assays and telomere lengths after multiple testing correction (Supplementary Table 12 and Supplementary Fig. 13), including some well-established, telomere-related proteins (for example, TK1, CDA and PARP1). For TK1, SPRED2 and BCL2L15, MR-Egger analysis highlighted the potential presence of pleiotropy, which might invalidate MR assumptions. One protein, RPA2, binds single-stranded DNA to protect from instability and breakage and recently has been shown to be involved in telomere maintenance[46]. The remaining associations were previously unreported and warrant future functional studies to elucidate the mechanism by which they mediate telomere length. Of the *trans* colocalizing proteins, 183 of 256 (71%) were found in the 12q24.12 locus containing *SH2B3*, which is known to be highly pleiotropic. Of the remaining *trans* colocalizing protein assay associations, six exhibited colocalization with more than one locus (Supplementary Fig. 14). These included FLT3LG for which *trans* pQTL signals colocalized with variants in *ATM*, *TERT* and *SETBP1* loci.

We also examined the overlap between our rare-variant telomere length analyses and rare pQTLs described in ref. 47. At the variant level, no germline overlapping variants were identified. At the gene level we identified one significant and two suggestive overlapping signals between a pQTL and PC1 telomere length. The significant association implicated a *trans* association between rare loss-of-function variants in *TERT* associated with shorter telomere length (ptvraredmg, $P = 1.7 × 10^{-134}$, $\beta = -0.52$ (−0.70 to −0.59)) and increased FLT3LG plasma abundance (ptvraredmg, $P = 4.68 × 10^{-9}$, $\beta = 0.52$ (0.35–0.69)). The remaining suggestive associations overlapped with *trans* pQTLs for α-fetoprotein (AFP) abundance, with putative loss of function for *ATM* and *ZNF451* being associated with shorter telomere length (ptvraredmg, $P = 5 × 10^{-27}$, $\beta = -0.15$ (−0.18 to −0.12)) and increased AFP (ptvraredmg, $P = 1.2 × 10^{-8}$ 0.25 (0.16–0.34)) and longer telomere length (ptv, $P = 1.1 × 10^{-11}$ 0.36 (0.25–0.46)) and decreased AFP (ptv, $P = 4.9 × 10^{-7}$, $\beta = -0.76$ (−1.1 to −0.47)), respectively.

### Causal gene prioritization

To prioritize putative causal genes in GWAS loci, we generated a list of 7,334 protein-coding genes overlapping a telomere-length PC1 locus and annotated this gene set with data integrated from seven separate sources (Supplementary Methods). Assuming equal weighting across all seven prioritization categories, we computed a simple sum to prioritize genes within each PC1 GWAS locus. Of the 7,334 protein-coding genes considered, 404 had a prioritization score >0 and a single gene was prioritized in 94 of the 192 PC1 telomere-length GWAS loci (Supplementary Tables 13 and 14). We found that these prioritized genes were more enriched ($P = 4.12 × 10^{-15}$) for the reactome pathway 'extension of telomeres' (R-HAS-180786) compared with 50 gene sets of the same size derived from randomly sampled closest genes ($P_{median} = 5.6 × 10^{-5}$) (Supplementary Fig. 15 and Supplementary Table 15).

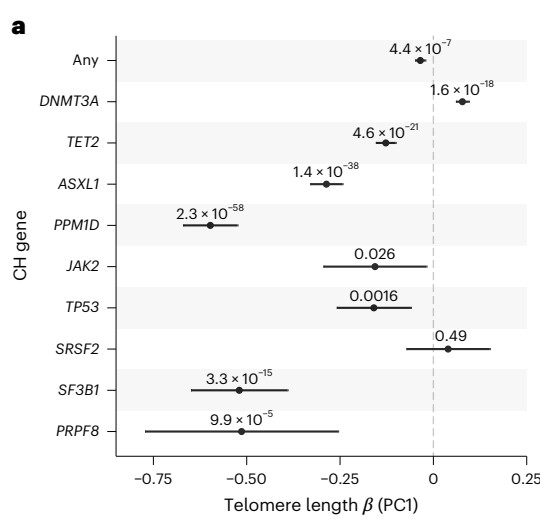
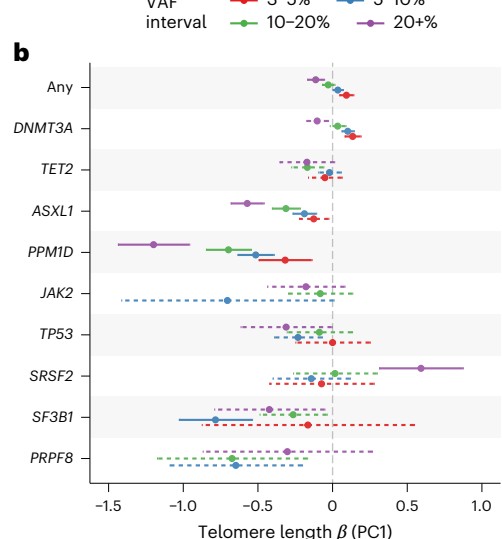

**Fig. 3 | Associations between telomere length and CH. a**, Collapsing analysis of somatic variants in select CH genes with telomere length PC1 metric. **b**, Collapsing analysis of somatic variants in CH genes stratified by VAF intervals (colors). Associations not reaching significance are shown with dashed error bars. In both plots, 'Any' indicates an overall analysis of the selected CH genes and estimates and 95% CIs (error bars) and P values (two-sided, unadjusted) are derived from fitting a linear model across 388,111 independent samples of broad NFE genetic ancestry.

## Multiancestry rare-variant analysis

Inclusion of individuals of non-European ancestries is critical for health equity and bolstering gene discovery[48,49]. Therefore, we performed additional GWAS, ExWAS and collapsing analysis on PC1 in five additional UKB genetic ancestry groups (admixed American/Hispanic (AMR), East Asian (EAS), South Asian (SAS), Ashkenazi Jewish (ASJ) and African (AFR); Supplementary Table 1). The broad genetic ancestry GWAS revealed a single locus in the AFR cohort that was not detected in the NFE cohort analyses (rs7577687, $P_{AFR} = 4.26 \times 10^{-8}$, $\beta_{AFR} = 0.11$ (0.07–0.15)) and there were no non-NFE genetic, ancestry-specific rare-variant associations, probably owing to the substantially smaller sample sizes of these populations in the UKB. A fixed-effect meta-analysis was then performed to combine results across ancestral strata, which detected an additional five loci (Supplementary Table 16). For the rare-variant ExWAS and collapsing meta-analysis, no additional study-wide significant genes were identified. However, there was a consistent improvement in observed statistical power, indicating that future cross-ancestry sequencing studies are likely to identify further causal gene telomere length associations (Extended Data Fig. 7).

## Telomere lengths in CH

Telomere length has been shown to be causally associated with clonal hematopoiesis (CH)[50,51]. In our rare-variant analyses, we identified several telomere length associations with five known CH driver genes (ExWAS: *CALR* and *JAK2*; collapsing: *CALR*, *TET2*, *ASXL1* and *PPM1D*) (Supplementary Tables 7 and 10), which we reasoned are probably driven by somatic events rather than germline inherited variation (Supplementary Fig. 10). To investigate this further, we performed somatic variant calling in 15 established CH and myeloid cancer driver genes (Supplementary Table 18) using the complementary UKB higher coverage exome sequencing data[52]. Using these somatic CH calls, and adjusting for age, sex and smoking status, we performed collapsing analyses with our PC1 metric and replicated the previously described association between overall CH and shorter telomere length[50,53] (Fig. 3a). By analysis of CH driver genes individually, we found that most followed the same pattern of association with shorter telomere length, including *SF3B1* ($P = 3.3 \times 10^{-15}$, $\beta = -0.52$ (−0.65 to −0.39)) and *PRPF8* ($P = 9.88 \times 10^{-5}$, $\beta = -0.51$ (−0.77 to −0.26)). Conversely, we discovered that CH driven by mutations in *DNMT3A* was significantly associated with longer telomere length ($P = 1.61 \times 10^{-18}$, $\beta = 0.08$ (0.06–0.10)) (Fig. 3a and Supplementary Table 18).

To investigate these associations further, and particularly to distinguish cause from effect in the context of telomere length measures ascertained from bulk blood, we performed subsequent analyses stratifying by the size of the mutant CH clone (Supplementary Table 19). Specifically, we reasoned that, in individuals with small CH clones (for example, VAF < 5%), most blood leukocytes would derive from wild-type (non-CH) cells and therefore reflect background telomere length. In comparison, in individuals with larger CH clones, average telomere length across blood cells would increasingly reflect telomere length within the mutant CH clone itself.

Small clones (for example, VAF 3–5%) were associated with longer telomere length for overall CH ($P = 1.05 \times 10^{-4}$, $\beta = 0.09$ (0.05–0.14)) and *DNMT3A*-mutant CH ($P = 8.6 \times 10^{-6}$, $\beta = 0.13$ (0.08–0.19)), consistent with previous reports that longer telomere length promotes CH acquisition (Fig. 3b and Supplementary Table 19)[20,50]. However, intriguingly, we discovered the inverse association for some other CH drivers, where small clones were associated with shorter telomere length, suggesting that acquisition of certain CH subtypes is promoted by shorter telomeres. A notable example was *PPM1D*, consistent with reports of high prevalence of *PPM1D*-mutant CH in individuals with inherited short telomere disorders[54,55].

Also aligning with previous reports, for CH overall and for most individual CH driver genes, we observed progressive shortening of telomere length with increasing clone size (any $P = 1.3 \times 10^{-14}$, $\beta = -0.49$ (−0.61 to −0.36)), probably reflecting accelerated telomere attrition with cell division in expanding clones (Supplementary Table 19). However, a striking exception to this pattern was observed in *SRSF2*-mutant CH, in which large clones were unexpectedly associated with longer telomere length ($P = 2.2 \times 10^{-6}$, $\beta = 1.36$ (0.81–1.91)), suggesting that *SRSF2* mutations may mediate telomere elongation in CH.

## Discussion

The present study of 462,666 multiancestry individuals presents a comprehensive, technically robust, genetic interrogation of telomere length. Importantly, we discovered that qPCR- and WGS-derived

estimates of telomere length capture additional genetic associations with telomere length. Combining these metrics via PCA not only enhanced downstream analyses, but also allowed us to discriminate artefactual signals (that is, associations with PC2). This has important implications for future population-based studies, because it suggests that, where possible, the most robust assessments should leverage both metrics.

Through both common and rare-variant-oriented studies, we described several telomere length loci that give insight into telomere biology. For example, we uncovered antagonistic allelic heterogeneity in *ACD* and *RTEL*1, highlighting the complex role for rare variants in telomere homeostasis and their role in disease. Moreover, the disease associations with both shorter and longer telomere length underscore the challenge of therapy development, where perturbation of balanced antagonistic effects might lead to important off-target effects. Integrative analysis of telomere length and proteomic data identified a number of putatively causal relationships, identifying known drug targets (for example, PARP1) and providing additional support for therapeutic modulation of nucleotide metabolism via TK1 and CDA[35]. We also identified a previously undescribed association between PTVs in *BRIP1* and *G3BP1* with shorter and longer telomere length, respectively. Although *BRIP1* is a helicase known to be involved in DNA damage response and *G3BP1* is involved in stress granule formation, their role in modulating telomere length is currently unclear and will require functional work in future studies.

Previous studies[11,50] have highlighted a causal, bidirectional relationship between telomere length and CH. In the present study, we uncovered driver gene-specific links between CH and telomere length, providing additional insights into the mechanisms driving clonal expansion. Longer telomeres predispose to *DNMT3A*-mutant CH, perhaps by extending cellular replicative potential, whereas this is not the case for some other CH driver genes, including *PPM1D*. It is notable that *PPM1D*-mutant CH is known to be particularly enriched among individuals with inherited short telomere disorders[54] and in individuals exposed to DNA-damaging chemotherapies that appear to shorten telomeres[56–58]. Taken together, we hypothesize that *PPM1D* mutations are specifically advantageous to blood stem cells in the context of critically short telomeres, perhaps by conferring resistance to the replicative senescence that would ordinarily occur in this setting.

It is also notable that mutations in particular splicing genes, such as *SRSF2*, have been shown to drive CH exclusively in older individuals[59], by which time telomeres have naturally shortened with age. The discovery that telomeres in *SRSF2*-mutant CH do not appear to shorten as clones expand, or even to elongate, contrasts starkly with the accelerated attrition of telomeres with clonal expansion driven by other CH genes. The possibility that *SRSF2* mutations confer advantage through telomere modulation offers one explanation for the expansion of these mutant clones specifically in older age; however, further functional studies are required to validate and elucidate the underlying biological mechanisms involved. In summary, our findings support a key role for telomere maintenance in the development of CH, via mechanisms specific to the mutant gene driving clonal expansion. As CH is a causal risk factor for progression to myeloid cancers and for a range of nonhematological diseases, with larger CH clones conferring higher risks[60,61], therapeutic modulation of telomere biology might be an important focus as strategies for prevention and treatment of CH and its sequelae.

## Online content

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

[1]Centre for Genomics Research, Discovery Sciences, BioPharmaceuticals R&D, AstraZeneca, Cambridge, UK. [2]Center for Genomics Research, Discovery Sciences, BioPharmaceuticals R&D, AstraZeneca, Waltham, MA, USA. [3]Biosciences COPD & IPF, Research and Early Development, Respiratory & Immunology, BioPharmaceuticals R&D, AstraZeneca, Gaithersburg, MD, USA. [4]Translational Science and Experimental Medicine, Research and Early Development, Respiratory & Immunology, BioPharmaceuticals R&D, AstraZeneca, Gothenburg, Sweden. [5]Translational Science and Experimental Medicine, Research and Early Development, Respiratory & Immunology, BioPharmaceuticals R&D, AstraZeneca, Cambridge, UK. [6]Department of Mathematical Sciences, Middle Tennessee State University, Murfreesboro, TN, USA. [7]Department of Cardiovascular Sciences, University of Leicester and Leicester NIHR Biomedical Research Centre, Leicester, UK. [8]Precision Medicine & Biosamples, Oncology R&D, AstraZeneca, Dublin, Ireland. [9]Department of Haematology, Cambridge University Hospitals NHS Foundation Trust, Cambridge, UK. [10]Department of Haematology, University of Cambridge, Cambridge, UK. [11]BioPharmaceuticals R&D, AstraZeneca, Cambridge, UK. [12]Department of Medicine, University of Melbourne, Austin Health, Melbourne, Victoria, Australia. [31]These authors contributed equally: Oliver S. Burren, Ryan S. Dhindsa, Sri V. V. Deevi. ✉e-mail: slav.petrovski@astrazeneca.com

## AstraZeneca Genomics Initiative

Rasmus Ågren[13], Lauren Anderson-Dring[1], Santosh Atanur[1], David Baker[14], Maria Belvisi[15], Mohammad Bohlooly-Y[16], Lisa Buvall[17], Sophia Cameron-Christie[1], Suzanne Cohen[18], Regina F. Danielson[19], Shikta Das[1], Andrew Davis[20], Guillermo del Angel[2], Sri V. V. Deevi[1,31], Wei Ding[21], Brian Dougherty[22], Zammy Fairhurst-Hunter[1], Manik Garg[1], Benjamin Georgi[4], Carmen Guerrero Rangel[1], Andrew Harper[1], Carolina Haefliger[1], Mårten Hammar[13], Richard N. Hanna[23], Pernille B. L. Hansen[18], Jennifer Harrow[1], Ian Henry[5], Sonja Hess[2], Ben Hollis[1], Fengyuan Hu[1], Xiao Jiang[1], Kousik Kundu[1], Zhongwu Lai[24], Mark Lal[18], Glenda Lassi[5], Yupu Liang[22], Margarida Lopes[1], Eagle Lou[25], Kieren Lythgow[1], Stewart MacArthur[1], Meeta Maisuria-Armer[1], Ruth March[8], Carla Martins[15], Dorota Matelska[1], Karine Megy[1], Rob Menzies[18], Erik Michaëlsson[26], Fiona Middleton[27], Bill Mowrey[22], Daniel Muthas[4], Abhishek Nag[1], Sean O'Dell[1], Erin Oerton[1], Yoichiro Ohne[19], Henric Olsson[4], Amanda O'Neill[1], Kristoffer Ostridge[4], Dirk Paul[1], Bram Prins[1], Benjamin Pullman[1], William Rae[1], Arwa Raies[1], Anna Reznichenko[13], Xavier Romero Ros[19], Hitesh Sanganee[21], Ben Sidders[28], Mike Snowden[21], Stasa Stankovic[1], Helen Stevens[1], Ioanna Tachmazidou[1], Haeyam Taiy[1], Lifeng Tian[25], Christina Underwood[14], Coralie Viollet[1], Anna Walentinsson[13], Lily Wang[25], Qing-Dong Wang[29], Eleanor Wheeler[1], Ahmet Zehir[30] & Zoe Zou[1]

[13]Translational Science and Experimental Medicine, Research and Early Development, Cardiovascular, Renal and Metabolism, BioPharmaceuticals R&D, AstraZeneca, Gothenburg, Sweden. [14]Bioscience Metabolism, Research and Early Development, Cardiovascular, Renal and Metabolism, BioPharmaceuticals R&D, AstraZeneca, Gothenburg, Sweden. [15]Research and Early Development, Respiratory and Immunology, BioPharmaceuticals R&D, AstraZeneca, Gothenburg, Sweden. [16]Centre for Genomics Research, Discovery Sciences, BioPharmaceuticals R&D, AstraZeneca, Gothenburg, Sweden. [17]Bioscience Renal, Research and Early Development, Cardiovascular, Renal and Metabolism, BioPharmaceuticals R&D, AstraZeneca, Gothenburg, Sweden. [18]Bioscience Asthma and Skin Immunity, Research and Early Development, Respiratory and Immunology, BioPharmaceuticals R&D, AstraZeneca, Cambridge, UK. [19]Research and Early Development, Cardiovascular, Renal and Metabolism, BioPharmaceuticals R&D, AstraZeneca, Gothenburg, Sweden. [20]Discovery Sciences, BioPharmaceuticals R&D, AstraZeneca, Cambridge, UK. [21]Alexion, AstraZeneca Rare Disease, Boston, MA, USA. [22]Early Oncology, Oncology R&D, AstraZeneca, Waltham, MA, USA. [23]Bioscience Immunology, Research and Early Development, Respiratory and Immunology, BioPharmaceuticals R&D, AstraZeneca, Cambridge, UK. [24]Oncology Data Science, Oncology R&D, AstraZeneca, Waltham, MA, USA. [25]Centre for Genomics Research, Discovery Sciences, BioPharmaceuticals R&D, AstraZeneca, Shanghai, China. [26]Early Clinical Development, Research and Early Development, Cardiovascular, Renal and Metabolism, BioPharmaceuticals R&D, AstraZeneca, Gothenburg, Sweden. [27]Business Development, AstraZeneca, Cambridge, UK. [28]Oncology Data Science, Oncology R&D, AstraZeneca, Cambridge, UK. [29]Bioscience Cardiovascular, Research and Early Development, Cardiovascular, Renal and Metabolism, BioPharmaceuticals R&D, AstraZeneca, Gothenburg, Sweden. [30]Precision Medicine and Biosamples, Oncology R&D, AstraZeneca, New York, NY, USA.

## Methods

### Ethics declarations

The protocols for the UKB are overseen by the UK Biobank Ethics Advisory Committee (EAC); for more information see https://www.ukbiobank.ac.uk/ethics and https://www.ukbiobank.ac.uk/wp-content/uploads/2011/05/EGF20082.pdf. Informed consent was obtained for all participants. The Northwest Research Ethics Committee reviewed and approved UKB's scientific protocol and operational procedures (REC reference no. 06/MRE08/65). Data for the present study were obtained and research conducted under the UKB application license nos. 24898 and 68574.

### WGS processing, QC and variant calling

WGS data of the UKB participants were generated by deCODE Genetics and the Wellcome Sanger Institute as part of a public–private partnership involving AstraZeneca, Amgen, GlaxoSmithKline, Johnson & Johnson, Wellcome Trust Sanger, UK Research and Innovation and the UKB. Sequencing was carried out in two centers (deCODE facility in Reykjavik, Iceland and the Wellcome Sanger Institute in Cambridge, UK). Genomic DNA underwent paired-end sequencing on Illumina NovaSeq6000 instruments with a read length of 2×151 and an average coverage of 32.5× (refs. [62,63]). Conversion of sequencing data in BCL format to FASTQ format and the assignments of paired-end sequence reads to samples were based on ten-base barcodes, using bcl2fastq (v.2.19.0). Initial QC was performed by deCODE and Wellcome Sanger, which included sex discordance, contamination, unresolved duplicate sequences and discordance with microarray genotyping data checks. From a total UKB cohort of 503,310 participants, 807 had withdrawn consent before WGS whereas 10,949 had no suitable sample for sequencing. The 50,0101 samples were sequenced as part of the Vanguard phase of the UKB WGS project such that, in total, 492,729 samples from 491,554 individuals were sequenced. After removing replicates, duplicates and an additional 91 individuals who withdrew consent after the sequencing had commenced, a total of 490,397 primary samples were available.

UKB genomes were processed at AstraZeneca using the provided CRAM format files. A custom-built Amazon Web Services cloud compute platform running Illumina DRAGEN Bio-IT Platform Germline Pipeline (v.3.7.8) was used to align the reads to the GRCh38 genome reference and to call small variants. Variants were annotated using SnpEff (v.4.3)[64] against Ensembl (Build 38.92)[65].

Finally, 490,348 (99.99%) sequences remained after removing contaminated sequences (verifybamid_freemix ≥ 0.04) using VerifyBAMID[66] or that had low CCDS coverage (<94.5% of CCDS r22 bases covered with ≥10-fold coverage).

### UKB WGS cohort definition

For the remaining 490,348 WGS samples. we used KING (v.2.2.3)[67] to identify individuals with first-degree relatives, which we then randomly pruned such that there were no pairs of samples with a kinship coefficient >0.354, to leave 490,216 (99.93%) WGS samples. We used peddy (0.4.2)[68] and 1000genomes data to classify individuals into broad genetic ancestries (peddy_prob ≥ 0.9) using the gnomAD classifier[69] to subdivide European (EUR) into NFE and ASJ individual broad genetic ancestries. We performed additional QC on NFE broad genetic ancestry samples using peddy-derived PCs, removing samples that fell outside 4 s.d. from the mean over the first four PCs. Next, we removed sex-discordant samples to leave 482,839 (98.4%) samples with TelSeq telomere length estimates (Extended Data Fig. 1). Final cohort sizes stratified by ancestry are described in Supplementary Table 1.

### WES

Full details of the WES and subsequent variant calling and annotation of the UKB cohort are described in Wang et al.[16]. Briefly, genomic DNA underwent paired-end 75-bp WES at Regeneron Pharmaceuticals using the IDT xGen v.1 capture kit on the NovaSeq6000 platform. Reads were aligned to GRCh38 and small indels (inserts and deletions) and SNVs called using running Illumina DRAGEN Bio-IT Platform Germline Pipeline v.3.0.7. The resultant catalog of variants was annotated using snpEFF v.4.3 (ref. [64]), Ensembl v.38.92 (ref. [65]), REVEL[70] and MTR[71] scores.

### Estimating telomere length from WGS data

We used TelSeq[10] v.0.0.2 to estimate telomere length using WGS data in the quality-controlled cohort of 482,839 UKB individuals. We used read length (−r) 150 and k-mer size (−k) 10 to match the proportional threshold (40%) for a read to be classified as of telomeric origin, as described in Ding et al.[10].

### Quantitative PCR telomere length estimates

For qPCR telomere length measurements, we used those available through UKB (field ID 22191) derived from baseline samples for a total of 472,518 participants. We used a rank inverse, normal transformed, relative telomer to single copy gene (T:S)-adjusted ratios without further adjustment given the extensive QC already performed on these measurements[72]. In total we identified 9,852 (2%) of the qPCR samples that lacked a matching TelSeq telomere length estimate from downstream analyses. We found that qPCR measurements in this set were significantly longer when compared with samples with both TelSeq and qPCR metrics (Student's $t$-test $P = 8.86 \times 10^{-42}$, two-sided, unadjusted, $\text{mean}_{\text{TelSeq\&qPCR}} = 5.0 \times 10^{-4}$, $\text{mean}_{\text{qPCR.only}} = 0.14$).

### Correcting TelSeq telomere length for technical confounders

We examined the correlation between inverse, normal rank transformed TelSeq- and UKB-adjusted qPCR T:S (UKB showcase field ID 22191) telomere length estimates finding modest agreement ($r^2 = 0.16$), perhaps indicating the presence of technical confounders. To mitigate this, for TelSeq telomere length estimates that were derived from WGS, we adapted the coverage correction method described in ref. [12]. Briefly, we used available Mosdepth[73] coverage files available across 482,839 WGS samples, which given the scale were calculated using a 'quantized' strategy that merges adjacent bases if they fall in the same coverage bin. Overall, four read depth bins were selected ((0–9), (10–19), (20–49) and (50+)). To compute overall coverage, we assumed that the coverage for a given base was the median of the read depth for that bin. As described in Taub et al.[12] we split the genome (GRCh38) into 1-kb tiles and removed those that overlapped regions with poor mappability, which were blacklisted overlapped known structural variants or were nonautosomal, resulting in 178,120 1-kb bins (approximately 6% of the genome). Then, for each sample we computed the average coverage across each bin. To facilitate downstream computation given the large size of the coverage matrix (that is, 482,839 × 178,120), we investigated the performance of randomly batching samples for coverage adjustment (Supplementary Note 2). This supported a strategy of 24 randomized batches (23 batches of 20,000 and 1 batch of 22,839 participants). For each batch we used a randomized PCA approach implemented in the R package 'rsvd' v.1.0.5 (ref. [74]) to estimate the first 300 PCs for each batch. To correct TelSeq telomere length estimates, we then fit a linear model (TelSeq$_{\text{raw}}$ ~ PC1–300), where '~' separates response and predictor variables, taking forward the resulting residuals as coverage-corrected TelSeq telomere length estimates.

To assess coverage-corrected TelSeq telomere length estimates we created Bland–Altman plots stratified by sex, ancestry and age using inverse normal transformed metrics for 462,666 participants in whom both metrics were available (Supplementary Figs. 2–5). Overall, we observed little bias in telomere length estimates when comparing qPCR and TelSeq methods. We used logistic regression to assess whether outlier status (by difference) was significantly associated with any of these biological metrics. Only AFR genetic ancestry was significantly associated with outlier status ($P = 1.16 \times 10^{-12}$, odds ratio

(OR) = 1.80, two-sided, unadjusted); however, when we added the rare *HBB*-coding variant carrier status into the model this association was significantly attenuated ($P = 2.65 \times 10^{-5}$, OR = 1.23) indicating that this might be driven by the genetic effects reported for qPCR telomere length within the *HBB* locus.

Finally, we looked for univariate association across 19 WGS sequence metrics (Supplementary Table 2) collected on each sample and both qPCR- and coverage-corrected telomere length estimates, using scaled WGS sequence metrics and inverse normal, rank transformed qPCR and coverage-adjusted telomere length estimates to facilitate comparison. We found that, overall, 14 and 16 WGS metrics were significantly associated with coverage-adjusted TelSeq and qPCR telomere length measurements (Extended Data Fig. 3 and Supplementary Table 3), respectively. Of these, many were highly correlated; however, we noted that a combination of coverage uniformity, total WGS reads and sequencing pipeline captured these, and so were included as covariates in downstream analyses (Supplementary Note 3). We also examined how mosaic loss of X or Y might differentially affect TelSeq and qPCR telomere length estimates, but did not find evidence for systematic differences between the two metrics (Supplementary Note 4).

In total 20,173 (4%) samples with TelSeq WGS telomere length estimates lacked matching qPCR estimates. There was no significant difference between TelSeq telomere length estimates in these samples compared with those with both telomere length measurements. A comparison of TelSeq measurements for this set and the set where both metrics were available did not detect a significant difference (Student's *t*-test $P = 0.17$ two-sided unadjusted, $\text{mean}_{\text{TelSeq\&qPCR}} = 5.0 \times 10^{-4}$, $\text{mean}_{\text{TelSeq.only}} = 7.0 \times 10^{-3}$).

## Correlation analysis

For the 462,666 samples that had telomere length estimates from both TelSeq and qPCR methods, we calculated the pairwise Pearson's correlation using the R 'cor' function. To assess the contribution and degree of collinearity between TelSeq and qPCR methods we fit the following model linear model using inverse rank, normal transformed age, TelSeq and qPCR (adjusted T:S ratio−UKB field ID 22191)

$$\text{Age} \sim \text{Telomere length}_{\text{TelSeq}} + \text{Telomere length}_{\text{qPCR}} + \text{Sex}.$$

We then used the R package olsrr (v.0.5.3) to compute variance inflation factors (VIFs) for each of the predictors, finding a mean VIF of 1.28 that indicated no evidence of collinearity. Overall removing telomere length$_{\text{TelSeq}}$ or telomere length$_{\text{qPCR}}$ from the model reduced $R^2$ by 0.01 and 0.02, respectively.

## PCA telomere length score

Across all 462,666 individuals with both telomere length measurements, we used the R built-in function 'prcomp' to combine the adjusted TelSeq and adjusted T:S ratio qPCR (UKB field ID 22191) inverse normal, transformed telomere length estimates. Each PCA consisted of two orthogonal principal axes with sample scores that were considered separate telomere length measurements or 'telomere length scores', with PC1 and PC2 explaining 77% and 23% of the variance, respectively (Fig. 1b). Overall PC1 was highly correlated with the standardized mean across TelSeq and qPCR metrics, whereas PC2 was correlated with their difference (Extended Data Fig. 4).

To assess performance for single and combined telomere length metrics we randomly sampled 10,000 participants from the full dataset. We used this training set to fit a simple linear model of a given telomere length metric with age (that is, age ~ telomere length$_{\text{metric}}$). Then, using the held-out participants, we used the model to predict age and assessed prediction performance as the root mean squared error of the age predictions. To perform crossvalidation and obtain CIs for these performance estimates, we performed this procedure 100×, sampling with replacement (Supplementary Fig. 6).

## NFE GWAS

We used UKB-imputed genotypes (UKB field ID 22828) to perform GWAS for qPCR, WGS, qPCR + WGS PC1 and qPCR + WGS PC2. Briefly, we performed additional QC, only taking forward NFE broad genetic ancestry samples with imputed genotypes (INFO > 0.7, MAC > 5) for which all telomere length metrics were available ($N = 438,351$). We used REGENIE (v.3.1)[14] with additional covariates of age, sex, genotyping plate, ancestry PCs 1−10 (as supplied by UKB) and WGS site. We excluded results for SNPs with the following (0.99 missingness, imputation INFO < 0.7 and p.Hardy−Weinberg equilibrium (p.HWE) > $1 \times 10^{-5}$). We found no evidence of genomic inflation (Supplementary Table 4). We selected sentinel SNPs and EUR-only broad genetic ancestry summary statistics from Codd et al.[8] for comparison (Supplementary Note 3 and Extended Data Fig. 5).

## LD score regression

We used ldsc (v.1.0.1)[15] to assess heritability and further assess possible stratification for each GWAS. Briefly, we used munge_stats.py on the cleaned summary stats (SNPs removed 0.95 missingness, imputation INFO < 0.4 and p.HWE > $1 \times 10^{-5}$), then used ldsc.py to estimate $h^2$ with the supplied 1000 Genomes LD score matrices.

## Defining GWAS loci

To define loci for each phenotype we selected significant variants ($P < 5 \times 10^{-8}$) and created regions ±1 Mb, creating a bespoke region (chr6: 25,500,000−34,000,000) for human leukocyte antigen. We then merged overlapping regions by phenotype, for each resultant region where the most significant variant was selected as the index; in the case of ties the variant closest to the middle of the region was selected. Finally we used the GenomicRanges[75] 'reduce' function to combine overlapping regions regardless of phenotype to define a set of nonredundant loci.

We used GCTA-COJO (v.1.94.1)[76] to perform stepwise model selection to define conditionally independent signals for each autosomal locus. Briefly, for each NFE GWAS we selected summary statistics for all variants (INFO ≥ 0.7) where $P < 1 \times 10^{-6}$. We then randomly sampled 50,000 individuals from the NFE cohort as the LD reference using BGENIX (v.1.1.7) and QCTOOLS (v.2.0.8)[77] to create bgen files for these individuals. Finally we used PLINK2 (ref. 78) to convert the resultant bgen files to binary PLINK 1.x format suitable for input into GCTA-COJO (gcta --cojo-slct) using default settings (--cojo-wind 10000; --cojo-p 5e-8; --cojo-collinear 0.9). For variants on the X chromosome we applied a similar approach but replaced 50,000 reference individuals with 50,000 randomly sampled female individuals of NFE ancestry and as a result of increased LD extended window size to 50 Mb (ref. 79).

To assess a list of previously reported loci, we compiled a list of significant ($P < 5 \times 10^{-8}$) variants from refs. 8,11,12 and the GWAS catalog[80] using the 'telomere length' term (EFO_0004505), downloaded on 11 July 2023. We then defined 2-Mb regions centered on each variant.

## Single causal variant fine-mapping

For variant fine-mapping, under the single causal variant we selected autosomal variants from NFE GWAS and divided these into approximately independent LD blocks using regions defined in ref. 81. We then used the single-variant fine-mapping[82,83] approach as implemented in https://github.com/ollyburren/rCOGS to assign 95% credible sets.

## SuSIE fine-mapping

We used SuSIE[84] to perform fine-mapping of all autosomal telomere length PC1 GWAS loci. Briefly, we selected a reference panel of 10,000 unrelated NFE broad genetic ancestry individuals for LD estimation. For each autosomal PC1 locus, we selected NFE telomere length summary stats and used PLINK to compute LD matrices across reference panel individuals. We then used the susie_rss function in the R package 'susieR' (v.0.12.35) to perform fine-mapping with $L = 10$, using the

susie_get_cs() function to obtain 95% credible sets (Supplementary Table 13).

## ExWAS

We carried out a virtual ExWAS of telomere length using WGS genotypes stratified by the broad genetic ancestry groupings: NFE ($n = 439{,}491$), SAS ($n = 9{,}349$), AFR ($n = 8{,}162$), EAS ($n = 2{,}362$), ASJ ($n = 1{,}201$) and AMR (675) ancestral groups. Briefly we selected unrelated individuals within each genetic ancestry stratum with telomere length and WGS data using the same method as described in UKB WGS cohort definition above. Finally, we removed individuals with a known hematological malignancy at sampling ($N_{overall} = 3{,}196$, NFE = 3,073, SAS = 42, AFR = 44, EAS = 9, AMR = 0). We took forward variants that passed the variant QC as described in Wang et al.[16] which had an MAC > 5. We used a linear model of the form telomere length$_{PC1}$ ~ genotype + age + sex + age2 + Peddy$_{PC1:4}$ + SequenceSite to assess the association of genotype with telomere length using the R 'PEACOK' package[16]. In the present study, genotype was coded as a genotypic (AA = 0, AB = 1, BB = 2), dominant (AA = 0, AB = 1, BB = 1) or recessive model (AA = 0, AB = 0, BB = 1), where A and B are the reference and alternative alleles. For the NFE ancestral group, we assessed 326,846, 326,846 and 62,716 variants for the dominant, genotypic and recessive models, respectively (carrier count ≥5). For the NFE analyses we reported the most significant model–variant pair such that variants $P \leq 1 \times 10^{-8}$ for PC1 and $P > 1 \times 10^{-8}$ for PC2 and MAF < 0.1%. For PC1 associated variants passing QC we reran association analyses for each variant conditional on other significant rare variants within 2 Mb to check for independence.

## Collapsing analysis

To assess the contribution of very rare variants we carried out a collapsing burden analysis stratified by broad genetic ancestral groups as per the ExWAS analysis, removing individuals diagnosed with a hematological malignancy at sampling, using the method described in ref. 16. Briefly, we aggregated qualifying variants based within the unit of a gene for each ancestral grouping and used these counts in a linear regression using the R 'PEACOK' (v.1.1.3) package with the same covariates as for the ExWAS. We defined ten qualifying variant tests (Supplementary Table 9) that include a synonymous model as an empirical control. We used the empirical modeling of the null distribution from Wang et al. to define a genome-wide significant threshold of $P < 1 \times 10^{-8}$. In total we assessed 18,930 genes across all 10 models. For NFE analyses we report best QV model–gene pair for which $P \leq 1 \times 10^{-8}$ for PC1 and $P > 1 \times 10^{-8}$ for PC2.

To assess the leverage of individual variants on collapsing analysis genome-wide significant hits we employed a LOO analysis. For each gene, and qualifying variant model, we reperformed collapsing analysis, leaving out one variant at a time. In this approach, variants with a large influence on the overall collapsing analysis, when excluded, result in a concomitant change in statistical significance (Supplementary Fig. 12).

## Multiancestry meta-analysis

We performed IVW meta-analysis for ExWAS and collapsing across NFE, SAS, AFR, EAS, ASJ and AMR broad genetic ancestry groupings for variants with a carrier count ≥5 within each grouping. In the context of rare variants, IVW can be unstable so we compared IVW meta-analysis P values with those generated from Stouffer's method, weighting each study by the square root of the sample size. We found that both approaches generated similar P values, indicating that IVW in this setting was stable even for rare variants.

For GWAS multiancestral analysis we used REGENIE with the approach described for the NFE ancestry group to generate GWAS summary statistics for SAS, AFR, EAS and AMR cohorts. We used the locus definition approach described earlier to define significant loci for each ancestral strata, considering the PC1 NFE ancestry telomere length

loci defined in Supplementary Table 5. For GWAS, we used METAL[85] to perform IVW meta-analyses across all ancestry strata. We selected significant variants ($P_{meta} < 5 \times 10^{-8}$), removing those that were present in a single broad genetic ancestry, using these to define loci and index variants as described earlier and assessing these for overlap with NFE loci.

## Proteogenomic colocalization analysis

We overlapped significant ($P < 1.7 \times 10^{-11}$) pQTLs for the 2,923 Olink protein assays reported in ref. 44 with PC1 telomere length loci to obtain 2,905 protein–telomere length loci pairs, harboring variants associated with both telomere length and one or more plasma protein abundances (Supplementary Table 11). To perform colocalization we extracted NFE GWAS summary statistics for all matching telomere length and pQTL (discovery + replication) variants in the locus. Given that both telomere length and pQTL GWAS were performed on inverse normal, rank transformed outcome variables, we assumed sdY = 1 and used 'coloc' (v.5.2.3)[45] to assess evidence for colocalization with the single-variant approximate Bayes' factor method using default priors. We defined 'strong' and 'weak' evidence for colocalization as (PP.H4.abf + PP.H3.abf) > 0.99 and (PP.H4.abf/PP.H3.abf) ≥ 5 and (PP.H4.abf + PP.H3.abf) > 0.90 and (PP.H4.abf/PP.H3.abf) ≥ 0, respectively and categorized colocalizations as *cis/trans* using the classifications provided in ref. 44 (within 1 Mb of the gene encoding the protein).

For *cis*-colocalizing signals ($n = 10$) where there was strong evidence for a shared causal variant between protein abundance and telomere length, we performed MR as implemented in the R package 'MendelianRandomization' (v.0.9.0)[86]. Briefly, For all variants (MAF > 1%) in a locus we performed clumping using PLINK[78] with a reference sample of 10,000 randomly sampled, unrelated NFE ancestry UKB samples (----indep-pairwise 100 kb 1 0.3), taking forward these pruned variants as instrumental pQTL variables. For MR, we used PLINK to compute correlation matrices for pruned variants at each locus. We then used the 'mr_allmethods' function to assess support for whether pQTL instruments were causally associated with telomere length across 'simple median', 'weighted median', 'IVW' and 'MR-Egger regression' methods. We took the median, across all four methods, using a multiple corrected P value ($P < 0.005$) as indicative of a putative causal relationship. Finally, we flagged results where the MR-Egger intercept term deviated from 0, indicating the presence of horizontal pleiotropy, which might invalidate underlying MR assumptions.

## CH analysis

To detect putative CH, we used the pipeline described in ref. 52. Briefly, using the same GRCh38 genome reference-aligned reads as for WES germline variant calling, we ran somatic variant calling with GATK's Mutect2 (v.4.2.2.0). After QC we focused on a set of 15 genes (Supplementary Table 17) exhibiting age-dependent prevalence for further analyses, including only PASS variant calls with 0.03 ≤ VAF ≤ 0.4 and allelic depth ≥3 across an annotated set of variants.

For the analysis, we considered four different VAF cutoffs (3–5%, >5–10%, >10–20% and >20%; Supplementary Table 17) across NFE ancestry individuals. In total, after excluding 3,585 individuals diagnosed with either a hematological malignancy predating sample collection or a lymphocyte count >5 × 10⁹ cells per liter, we took forward 435,525 individuals for analysis. For overall CH driver subtype association (as shown in Fig. 1a), we fit a linear model telomere length$_{PC1}$ ~ (CH$_{VAF>0.03}$ + age + sex + age)/(sex + age2 + ancestry PC1:4 + ever-smoked + pack-years), where telomere length$_{PC1}$ represents the PC1 telomere length estimate and CH the carrier status for a particular CH driver subtype with VAF > 3%. We then repeated this analysis stratifying by nonoverlapping VAF cutoffs for each CH driver subtype. Finally, to get an overall association statistic between telomere length and VAF stratified by CH driver subtype, we repeated this analysis recoding each CH driver gene carrier status by VAF as an ordinal variable.

## Statistics and reproducibility

Except where specific software packages are named, all statistical analyses and plotting were conducted using R (v.4.1.0). No statistical methods were used to predetermine sample size.

## Reporting summary

Further information on research design is available in the Nature Portfolio Reporting Summary linked to this article.

## Data availability

Full summary association statistics generated in the present study will be made publicly available through our AstraZeneca CGR phenotype-WAS (PheWAS) Portal (http://ftp.ebi.ac.uk/pub/databases/gwas/summary_statistics/GCST90435001-GCST90436000/) or the GWAS catalog (GCST90435144 and GCST90435145). All WGS and qPCR data described in the present study are publicly available to registered researchers through the UKB data access protocol. Genomes can be found in the UKB showcase portal: https://biobank.ndph.ox.ac.uk/showcase/label.cgi?id=100314. The qPCR-derived telomere length estimates are available at https://biobank.ndph.ox.ac.uk/ukb/label.cgi?id=265 and WGS TelSeq estimates will be made available as a 'returned dataset'. Additional information about registration for access to the data is available at http://www.ukbiobank.ac.uk/register-apply. Data for the present study were obtained under resource application nos. 26041 and 68601.

## Code availability

Code supporting the present study is available from Zenodo via https://doi.org/10.5281/zenodo.12684065 (ref. 87). PheWAS and ExWAS association tests were performed using a customized framework, PEACOK (1.0.7). PEACOK is available on GitHub: https://github.com/astrazeneca-cgr-publications/PEACOK. In addition to the R packages mentioned in the text, we used pacman (v.0.5.1), data.table (v.1.14.0), magrittr (v.2.03), tidyverse (v.2.0.0), rtracklayer (v.1.54.0), GenomicRanges (v.1.46.1), cowplot (v.1.1.3), patchwork (v.1.2.0), biomaRt (v.2.5.3), ggrepel (v.0.9.5) and ggplot2 (v.3.4.4).

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

## Acknowledgements

The PCR-based measurement of telomere length was conducted using the UKB Resource under application no. 6077 and was funded by the UK Medical Research Council (MRC), Biotechnology and Biological Sciences Research Council and British Heart Foundation through the MRC (grant no. MR/M012816/1). V.C., C.P.N. and N.J.S. are supported by the National Institute for Health and Care Research, Leicester Cardiovascular Biomedical Research Centre (grant no. BRC-1215-20010). We thank the participants and investigators of the UKB study who made this work possible (resource application nos. 26041 and 65851). We also thank the AstraZeneca Centre for Genomics Research (CGR) Analytics and Informatics team for processing and analysis of sequencing data.

## Author contributions

O.S.B., R.S.D., Q. Wang and S.P. designed the study. O.S.B., R.S.D., S.V.V.D., S. Wen, A.N., J.M., F.H., K.R.S., Q. Wu and Q. Wang performed statistical analyses and data interpretation. S.V.V.D. and S. Wasilewski performed bioinformatic processing. D.S.L., N.R., V.C., C.P.N., N.J.S, M.F. and S.P. performed data interpretation. O.S.B. generated the figures and tables. O.S.B., R.S.D., M.F. and S.P. wrote the manuscript. O.S.B., R.S.D., S.V.V.D., S. Wen, A.N., J.M., F.H., K.R.S., S. Wasilewski, D.S.L., N.R., V.C., C.P.N., N.J.S., M.F., H.O., A.P., D.V., R.M., K.C., M.P., Q. Wang and S.P. reviewed and edited the manuscript.

## Competing interests

O.S.B., R.S.D., S.V.V.D., S. Wen, A.N., J.M., F.H., D.S.L., K.R.S., N.R., H.O., A.P., P.V., Q. Wu, R.E.M., S. Wasilewski, K.C. M.F., Q. Wang, M.N.P. and S.P. are current employees and/or stockholders of AstraZeneca. All other authors declare no competing interests.

## Additional information

**Extended data** is available for this paper at https://doi.org/10.1038/s41588-024-01884-7.

**Correspondence and requests for materials** should be addressed to Slavé Petrovski.

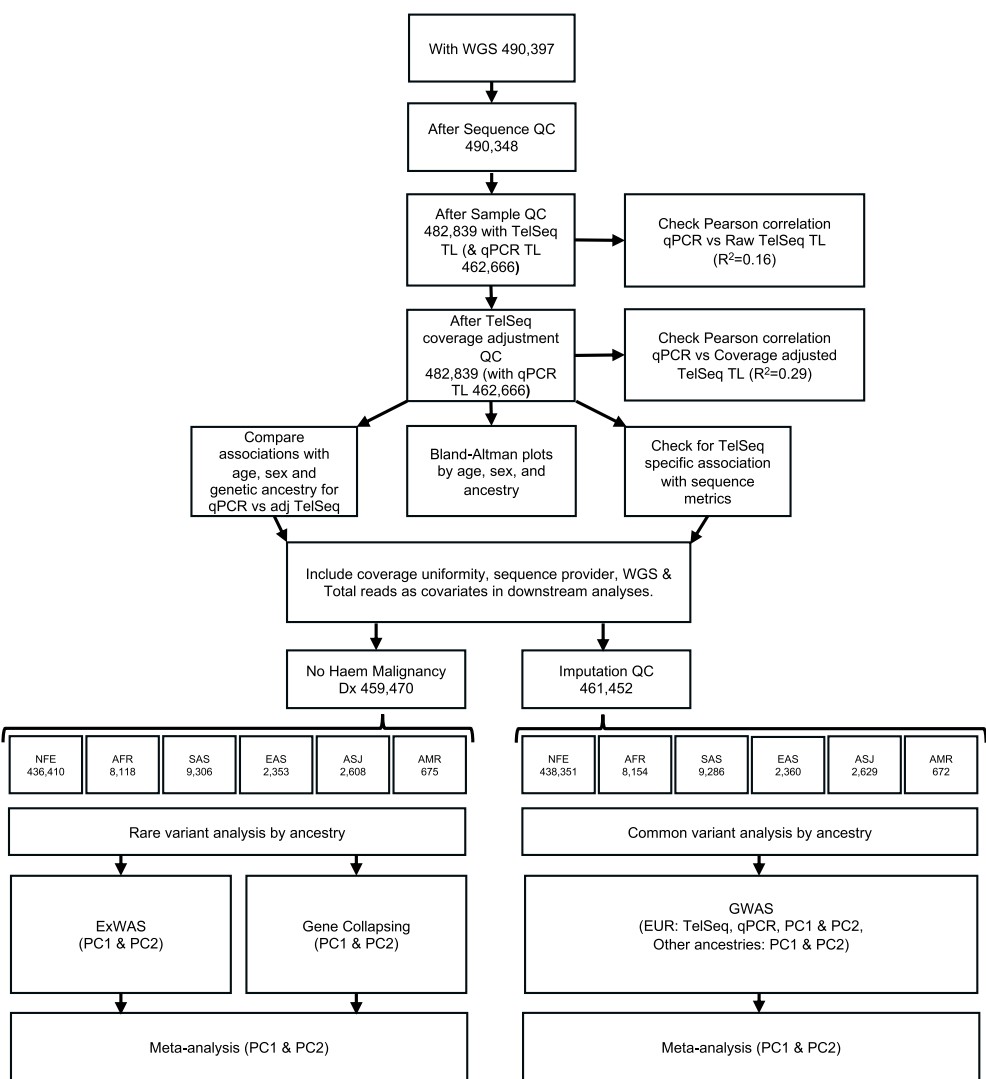

**Extended Data Fig. 1 | Flowchart of sample QC and analyses.** Abbreviations; WGS = Whole genome sequencing, Dx = Diagnosis, Broad genetic ancestry groupings - AFR = African, AMR = Admixed American/Hispanic ASJ = Ashkenazi Jewish, EAS = East Asian, NFE = Non-Finnish European, SAS = South Asian.

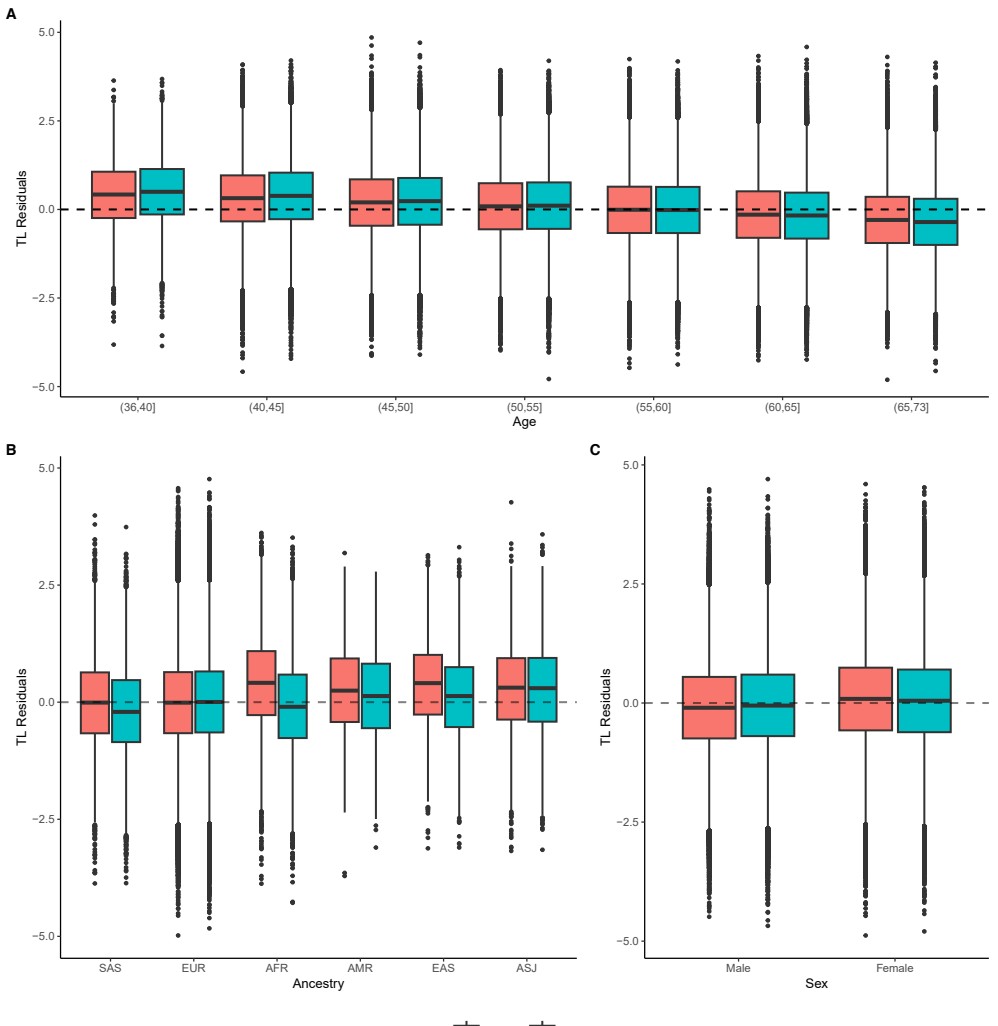

**Extended Data Fig. 2 | Age, ancestry, and sex relationships with TelSeq & qPCR telomere length measurements.** For each panel y-axes denote telomere length residuals after regressing out age, sex, or ancestry depending on the x-axis variable. In all panels N for qPCR and TelSeq + Coverage is 462,666 and 482,839 independent UKB participants respectively. For each boxplot the centre is the median, the lower and upper hinges indicate the 25th and 75th percentile and outliers are represented as individual points. (**A**) Boxplot of age by telomere length residuals. (**B**) Boxplot for broad genetic ancestry group (AFR = African, AMR = Admixed American/Hispanic ASJ = Ashkenazi Jewish, EAS = East Asian, NFE = Non-Finnish European, SAS = South Asian) by telomere length residuals. (**C**) Boxplot for sex by telomere length residuals.

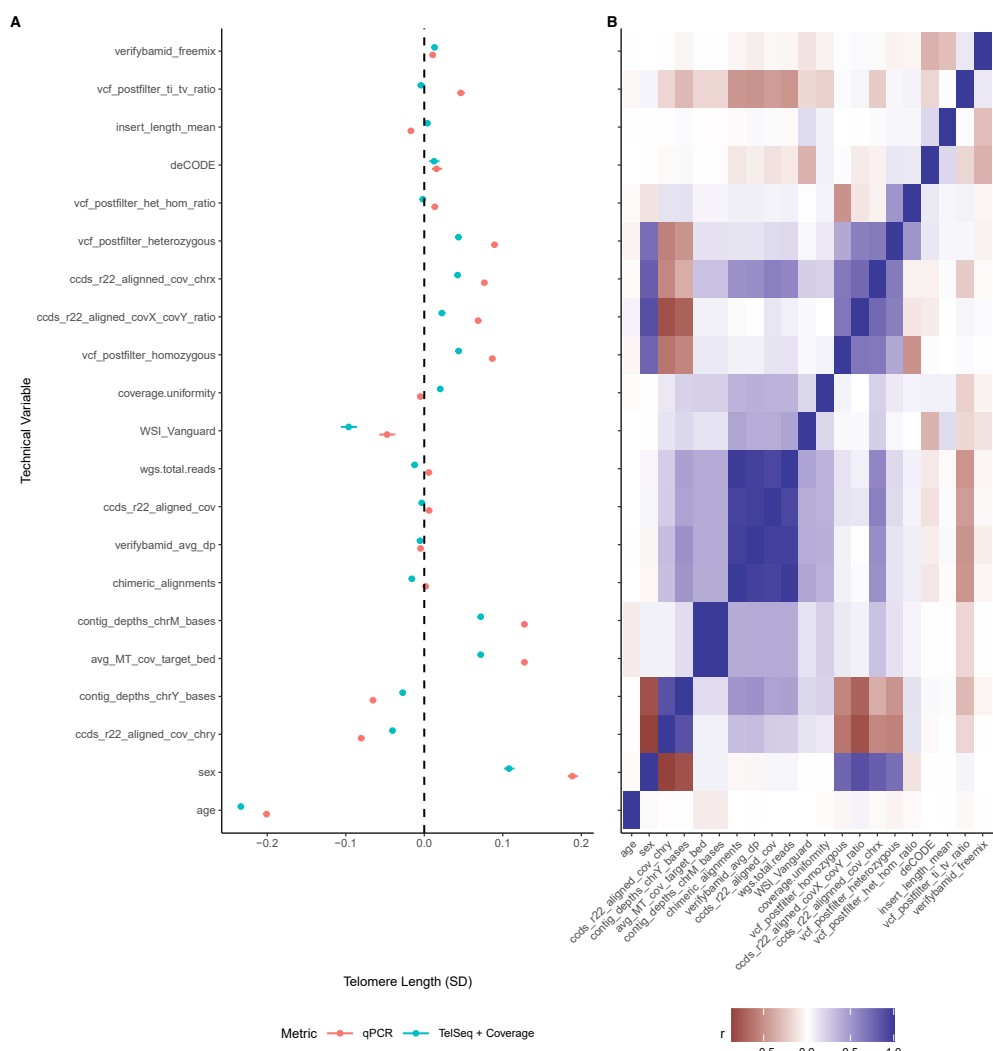

**Extended Data Fig. 3 | Association of whole genome sequencing technical variables with qPCR and coverage adjusted TL metrics. (A)** Forest plot of Bonferroni significant associations (P < 1 x 10-3) from a univariate linear regression of technical variables (two-sided) with either qPCR (coral) or inverse rank normal transformed TelSeq coverage adjusted (azure) telomere lengths (n = 462,666 independent samples). All variables have been standardised to facilitate effect size comparison on telomere length (x-axis), 95% confidence intervals are shown. A full table of all results with descriptions is available as Supplementary Table 3. Sequencing pipeline (deCODE, WSI, and WSI_vanguard (baseline)) and sex are treated as categorical variables. **(B)** Pearson correlation heatmap of significantly associated WGS technical variables. Variable order is derived from hierarchical (complete linkage) clustering of the full correlation matrix. Age and sex are included as biological variables with known associations with telomere length for comparison.

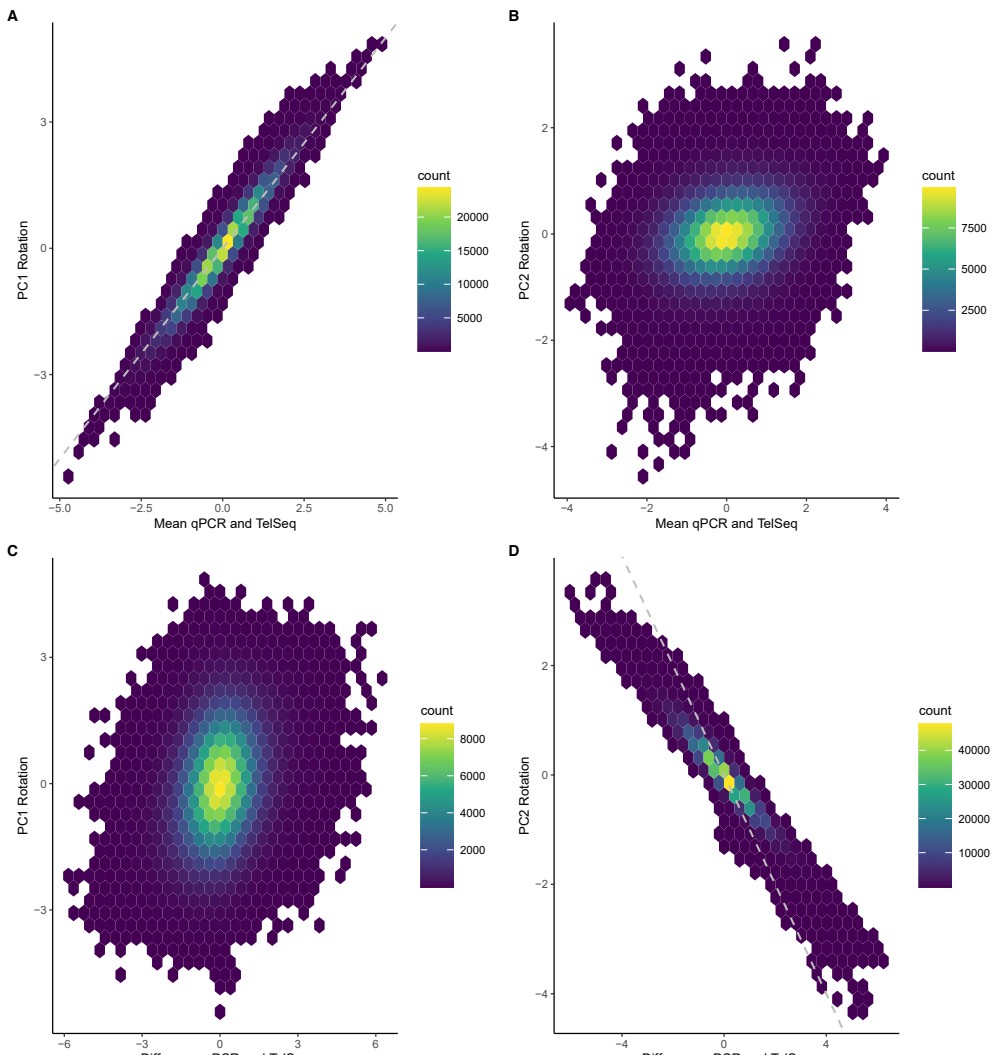

**Extended Data Fig. 4 | Comparison of PC1 and PC2 rotations.** Density plots of mean qPCR and TelSeq transformed telomere length estimates vs PC1 rotation values (**A**), mean qPCR and TelSeq transformed telomere length estimates vs PC2 rotation values (**B**), difference between qPCR and TelSeq transformed telomere length estimates vs PC1 rotation values (**C**), and difference between qPCR and TelSeq transformed telomere length estimates vs PC2 rotation values (**D**). Dotted lines indicate x = y (Top) and x = -y (Bottom) and are included for reference.

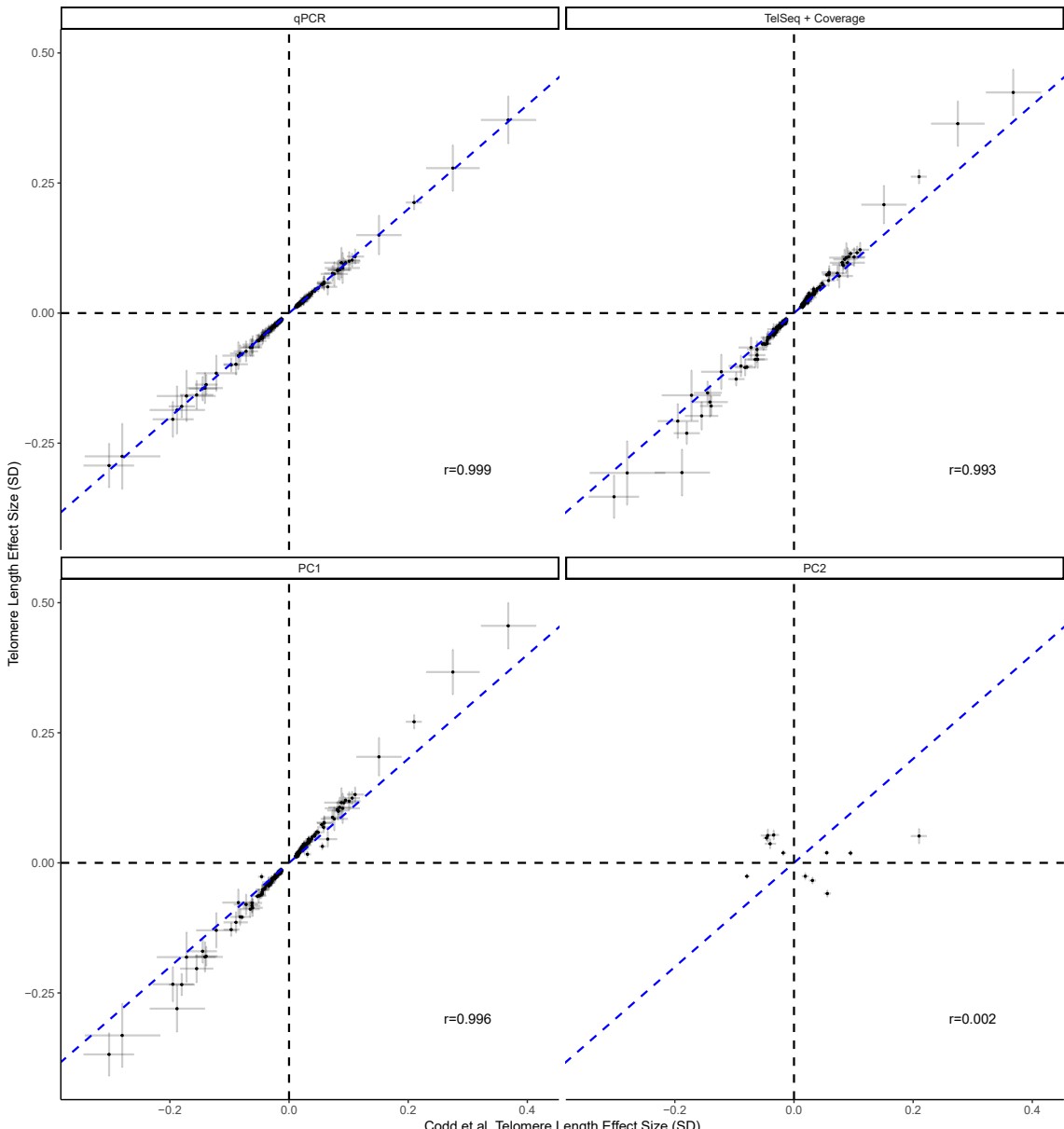

**Extended Data Fig. 5 | Comparison of GWAS effect sizes with *Codd* et al.** (y-axis) for different TL measurements effect sizes with EUR only effect sizes from *Codd* et al. (x-axis) and NFE from this study (P < 5 ×10⁻⁸). P values are derived from linear regression and are two sided and unadjusted. Crosses indicate 95% confidence intervals for each estimated effect size; Pearson's correlation coefficients are labelled on each panel; blue dotted line shows equivalence (x = y).

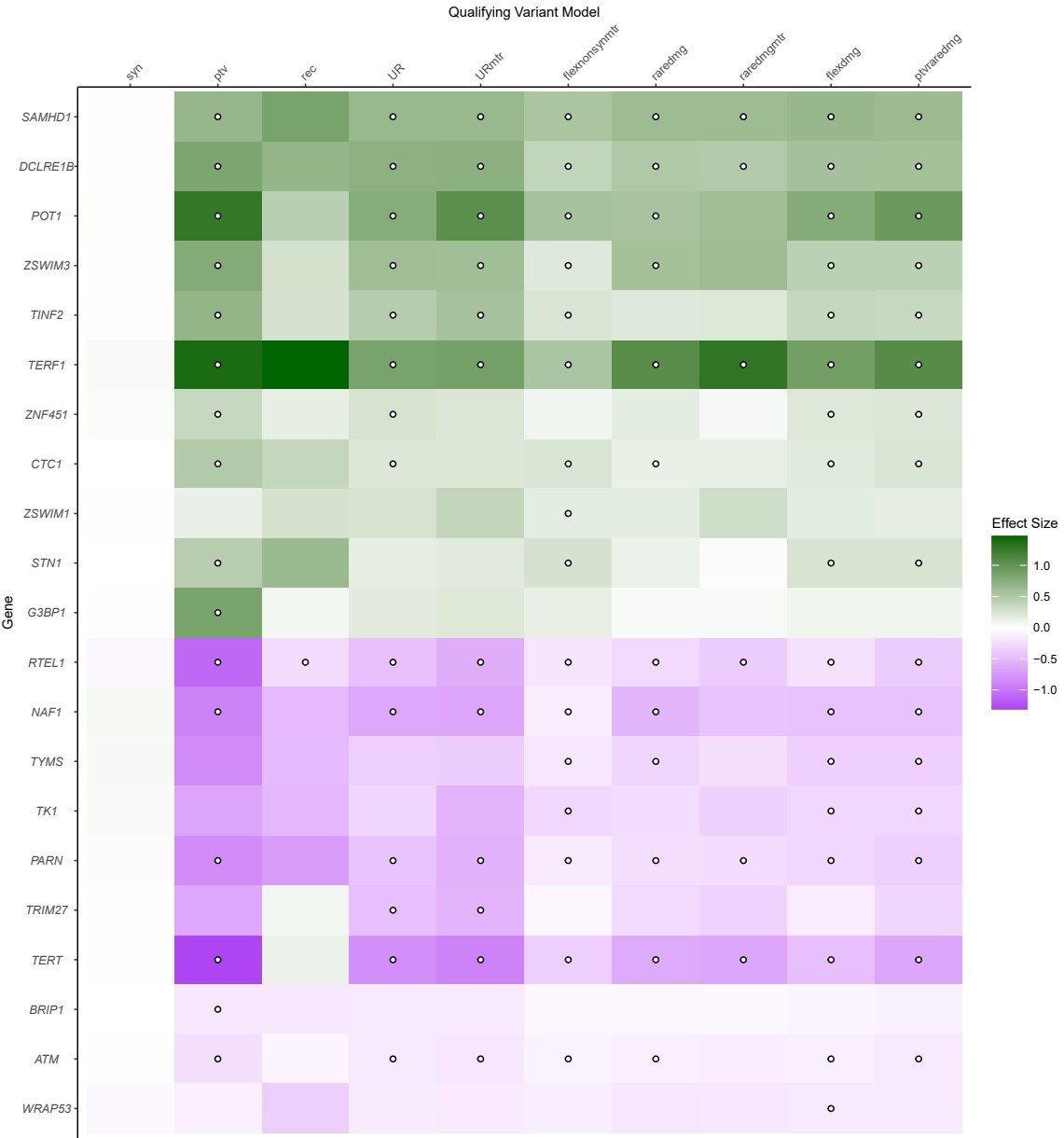

**Extended Data Fig. 6 | Heatmap of genome-wide significant telomere length associated genes from gene collapsing analyses.** Shading indicates effect size (green = unit increased telomere length, purple = unit decreased telomere length), points indicate genome-wide significance (P ≤ 1x10$^{-8}$). The x-axis indicates the different qualifying variant models implemented which are described fully in Wang et al. Briefly, ptv= rare protein truncating variants, UR = ultra rare variants, URmtr = ultra rare variants in missense intolerant regions

(MTR), raredmg = rare damaging (REVEL) variants, raredmgmtr = as raredmg but with additional MTR filter, flexdmg = flexible non-synonymous, flexnonsynmtr = as flexdmg but with additional MRT filter, ptvraredmg = Union of ptv and raredmg models, rec = recessive model, syn = synonymous variants (negative control). P values are derived from linear regression and are two sided and unadjusted.

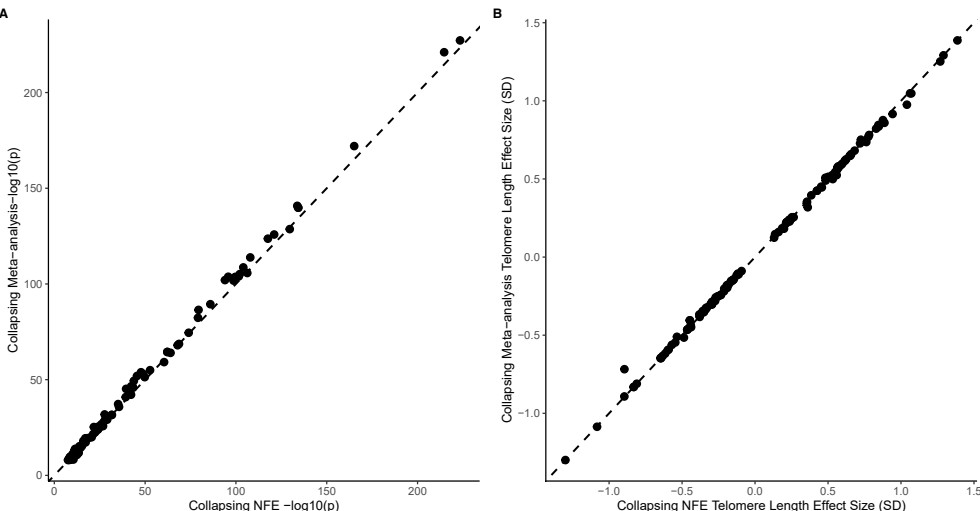

**Extended Data Fig. 7 | Comparison of p values for NFE and fixed effect cross-ancestry meta-analysis collapsing analysis (AFR (n = 8,154), ASJ (n = 2,629), EAS (n = 2,360), NFE (438,351) & SAS (9,286)) for the PC1 telomere length metric.** Only variants pNFE < 5 x 10$^{-5}$ for PC1 are shown, (**A**) significance -log10(P) (**B**) Telomere length effect size (SD). P-values were derived from inverse-weighted meta-analysis and are two-sided.

# Reporting Summary

## Statistics

For all statistical analyses, confirm that the following items are present in the figure legend, table legend, main text, or Methods section.

| n/a | Confirmed | |
|---|---|---|
| ☐ | ☒ | The exact sample size (*n*) for each experimental group/condition, given as a discrete number and unit of measurement |
| ☐ | ☒ | A statement on whether measurements were taken from distinct samples or whether the same sample was measured repeatedly |
| ☐ | ☒ | The statistical test(s) used AND whether they are one- or two-sided<br>*Only common tests should be described solely by name; describe more complex techniques in the Methods section.* |
| ☐ | ☒ | A description of all covariates tested |
| ☐ | ☒ | A description of any assumptions or corrections, such as tests of normality and adjustment for multiple comparisons |
| ☐ | ☒ | A full description of the statistical parameters including central tendency (e.g. means) or other basic estimates (e.g. regression coefficient) AND variation (e.g. standard deviation) or associated estimates of uncertainty (e.g. confidence intervals) |
| ☐ | ☒ | For null hypothesis testing, the test statistic (e.g. *F*, *t*, *r*) with confidence intervals, effect sizes, degrees of freedom and *P* value noted<br>*Give P values as exact values whenever suitable.* |
| ☒ | ☐ | For Bayesian analysis, information on the choice of priors and Markov chain Monte Carlo settings |
| ☒ | ☐ | For hierarchical and complex designs, identification of the appropriate level for tests and full reporting of outcomes |
| ☐ | ☒ | Estimates of effect sizes (e.g. Cohen's *d*, Pearson's *r*), indicating how they were calculated |

*Our web collection on statistics for biologists contains articles on many of the points above.*

## Software and code

Policy information about availability of computer code

| Data collection | Single-sample processing, on Amazon Web Services (AWS) cloud compute platform.<br>* Conversion of sequencing data in BCL format to FASTQ format and the assignments of paired-end sequence reads to samples based on 10-base barcodes; bcl2fastq v2.19.0 https:// support.illumina.com/sequencing/sequencing_software/bcl2fastq-conversion-software.html<br>* read alignment and variant calling performed on Illumina DRAG EN Bio-IT Platform Germline Pipeline v3.0.7 to align the reads to the GRCh38 genome reference [http://ftp.1000genomes.ebi.ac.uk/vol1/ftp/technical/reference/GRCh38_reference_genome/.] and perform small variant SNV and indel calling. SNVs and indels were annotated using SnpEFF v4.3 against Ensembl Build 38.92. We further annotated all variants with their gnomAD minor allele frequencies (gnomAD v2.l.1 mapped to GRCh38).<br>* For ancestry, we used PEDDY v0.4.2 with the ancestry labeled 1K Genomes Project reference sequence data for genetic ancestry predictions.<br>* For relatedness, we used ukb_gen_samples_to_remove() function from the R package ukbtools v0.11.3. |
|---|---|
| Data analysis | * WGS Telomere length estimation was performed using TelSeq v0.0.2 (https://github.com/zd1/telseq)<br>* PheWAS and exWAS association tests were performed using a custom built frame PEACOK (PEACOK 1.0.7), which is an extension and enhancement of PHESANT. PEACOK 1.0.7 can be found: https://github.com/astrazeneca-cgr-publications/PEACOK/versions/1.0.7<br>* GWAS was performed using REGENIE v3.1 (https://rgcgithub.github.io/regenie/)<br>* LD score regression was performed using LDSC v1.01 (https://github.com/bulik/ldsc)<br>* Approximate conditional association was performed using GCTA/COJO v1.94.1 (https://yanglab.westlake.edu.cn/software/gcta/#Download)<br>* Genotype data management and LD pruning was performed using PLINK v1.9 (https://www.cog-genomics.org/plink/) and PLINK v2.0 (https://www.cog-genomics.org/plink/2.0/)<br>* To call somatic CH variants we used Mutect2 v.4.2.2 (https://gatk.broadinstitute.org/hc/en-us/articles/4405443657499-Mutect2)<br>* Large-scale compute was done using AWS Batch computing environment. |

* We used genome sequence-derived genotypes for biallelic autosomal SNVs located in coding regions as input to the kinship algorithm included in KING v2.2.3.
* We use PLINK1 (v1.90b6.21) and PLINK2 (v2.00) for genotype processing and LD pruning.
* MAGMA v1.08 to integrate functional data to prioritise putative causal genes.
* PoPS v 0.2 to integrate functional data to prioritise putative causal genes.
* susieR v 0.12.35 library to perform finemapping
* QCTOOLS v2.0.6 and BCTOOLS v1.11 to manage genotype data
* Various downstream analysis and summarization were performed using R v4.1.0 https://cran.r-project.org. R library  MASS (7.3-51.6), pacman (0.5.1), data.table (v 1.14.0) tidyverse (2.0.0) ggplot2 (v3.4.4) rtracklayer (1.54.0), GenomicRanges (1.46.1), cowplot (1.1.3), patchwork (1.2.0), biomaRt (2.5.3), ggrepel (0.9.5) and ukbtools (v0.11.3)

For manuscripts utilizing custom algorithms or software that are central to the research but not yet described in published literature, software must be made available to editors and reviewers. We strongly encourage code deposition in a community repository (e.g. GitHub). See the Nature Portfolio guidelines for submitting code & software for further information.

## Data

Policy information about availability of data

All manuscripts must include a data availability statement. This statement should provide the following information, where applicable:
- Accession codes, unique identifiers, or web links for publicly available datasets
- A description of any restrictions on data availability
- For clinical datasets or third party data, please ensure that the statement adheres to our policy

Full summary association statistics generated in this study will be publicly available through our AstraZeneca Centre for Genomics Research (CGR) PheWAS Portal (http://azphewas.com/) or GWAS catalog (https://www.ebi.ac.uk/gwas/) [GCST90435144 & GCST90435145]. All whole-genome sequencing data and qPCR data described in this paper are publicly available to registered researchers through the UKB data access protocol. Genomes can be found in the UKB showcase portal: https://biobank.ndph.ox.ac.uk/showcase/label.cgi?id=100314. qPCR-derived TL estimates are available at https://biobank.ndph.ox.ac.uk/ukb/label.cgi?id=265, and WGS TelSeq estimates will be made available as a 'Returned Dataset' . Additional information about registration for access to the data is available at http://www.ukbiobank.ac.uk/register-apply/. Data for this study were obtained under Resource Application Numbers 26041 and 68601.

## Research involving human participants, their data, or biological material

Policy information about studies with human participants or human data. See also policy information about sex, gender (identity/presentation), and sexual orientation and race, ethnicity and racism.

| | |
|---|---|
| Reporting on sex and gender | All analyses included males and females. We report that sex was used as a covariate in the association analyses. |
| Reporting on race, ethnicity, or other socially relevant groupings | 94% of the cohort is of European ancestry. |
| Population characteristics | The average age was 57, and 54% of the cohort was female |
| Recruitment | Participants were recruited to the UK Biobank on a voluntary basis. Approx 500K individuals 40-69 years of age in 2006-2010 volunteered. Informed consent was obtained for all participants. It has previously been observed that participants are less likely to live in socioeconomically deprived areas than non-participants, and they tend to be healthier than non-participants, which may impact some of the reporting rates in comparison to what could be observed through random sampling from the UK population. Fry et al (10.1093/aje/kwx246). |
| Ethics oversight | The protocols for UK Biobank are overseen by The UK Biobank Ethics Advisory Committee (EAC), for more information see https://www.ukbiobank.ac.uk/ethics/ and https://www.ukbiobank.ac.uk/wp-content/up1oads/2011/05/EGF20082.pdf |

Note that full information on the approval of the study protocol must also be provided in the manuscript.

# Field-specific reporting

Please select the one below that is the best fit for your research. If you are not sure, read the appropriate sections before making your selection.

☒ Life sciences          ☐ Behavioural & social sciences          ☐ Ecological, evolutionary & environmental sciences

For a reference copy of the document with all sections, see nature.com/documents/nr-reporting-summary-flat.pdf

# Life sciences study design

All studies must disclose on these points even when the disclosure is negative.

| | |
|---|---|
| Sample size | There were 490,560 UKB participants with WGS data. We further subset the cohort based on QC metrics as described in the manuscript. No sample size calculations for power were performed. |

| | |
|---|---|
| Data exclusions | At the sample level, we excluded samples based on predefined exclusion criteria as detailed in the manuscript. Briefly, we excluded those that did not pass sequencing quality control thresholds. |
| Replication | We replicated the signals from our GWAS with signals from a prior GWAS performed on the same cohort (see Supplementary Note) and was not independent. |
| Randomization | This study is observational. Randomization was not applicable to this study. |
| Blinding | This study is observational, using coded de-identified data. Blinding was not applicable to this study. |

# Reporting for specific materials, systems and methods

We require information from authors about some types of materials, experimental systems and methods used in many studies. Here, indicate whether each material, system or method listed is relevant to your study. If you are not sure if a list item applies to your research, read the appropriate section before selecting a response.

## Materials & experimental systems

| n/a | Involved in the study |
|---|---|
| ☒ | ☐ Antibodies |
| ☒ | ☐ Eukaryotic cell lines |
| ☒ | ☐ Palaeontology and archaeology |
| ☒ | ☐ Animals and other organisms |
| ☒ | ☐ Clinical data |
| ☒ | ☐ Dual use research of concern |
| ☒ | ☐ Plants |

## Methods

| n/a | Involved in the study |
|---|---|
| ☒ | ☐ ChIP-seq |
| ☒ | ☐ Flow cytometry |
| ☒ | ☐ MRI-based neuroimaging |

