## [Peer Review File · Nature Genetics]

Peer Review Information

Manuscript Title: Genetic architecture of telomere length in 462,666 UK Biobank whole-genome sequences

Corresponding author name(s): Dr Slavé Petrovski

Reviewer Comments & Decisions:

Decision Letter, initial version:
--

22nd Oct 2023

Dear Slave,

Your Article, "Genetic architecture of telomere length in 462,675 UK Biobank whole-genome sequences" has now been seen by 3 referees. You will see from their comments copied below that while they find your work of considerable potential interest, they have raised quite substantial concerns that must be addressed. In light of these comments, we cannot accept the manuscript for publication, but would be very interested in considering a revised version that addresses these serious concerns.

In brief, while the reviews acknowledge the potential interest in and advance of your study of telomere length combining these two qPCR+WGS based measurements, there are a range of fundamental issues raised that require resolution.

Reviewer #1 thinks that your PC-based approach to combine qPCR and TelSeq has merit, but their specific comments are focused on a single overriding concern: the low correlation between qPCR and WGS LTL measurements, which needs thorough investigation and QC. They do provide specific instruction to do so.

Referee #2 is the most skeptical of the three: they suggest that the novelty and advance of your study falls short, given that many of your findings have already been reported in the Mendelian disorder literature.

Reviewer #3 strikes a similar overall tone to #1, sounding supportive but also making a number of useful suggestions for improvement. Notably, they also ask for further investigation of the differences between the qPCR and WGS measurements of LTL.

In our reading of these reviews, we think there is a path to publication and we do not think any of the specific requests made are unreasonable or impractical. We would highlight the concern regarding the correlation of the two LTL measurements as an especially important one, given these data form the

basis of the entire work. We would also recommend you consider how to improve the novelty in light of Reviewer #2's comments; we acknowledge they have not provided much specific guidance for such, but there are multiple standard post-GWAS analyses that could potentially be added, for example.

We hope you will find the referees' comments useful as you decide how to proceed. If you wish to submit a substantially revised manuscript, please bear in mind that we will be reluctant to approach the referees again in the absence of major revisions.

To guide the scope of the revisions, the editors discuss the referee reports in detail within the team, including with the chief editor, with a view to identifying key priorities that should be addressed in revision and sometimes overruling referee requests that are deemed beyond the scope of the current study. We hope that you will find the prioritised set of referee points to be useful when revising your study. Please do not hesitate to get in touch if you would like to discuss these issues further.

If you choose to revise your manuscript taking into account all reviewer and editor comments, please highlight all changes in the manuscript text file. At this stage we will need you to upload a copy of the manuscript in MS Word .docx or similar editable format.

*2) If you have not done so already please begin to revise your manuscript so that it conforms to our Article format instructions, available here. Refer also to any guidelines provided in this letter.

Please be aware of our guidelines on digital image standards.

[redacted]

If you wish to submit a suitably revised manuscript we would hope to receive it within 6 months. If you cannot send it within this time, please let us know. We will be happy to consider your revision so long as nothing similar has been accepted for publication at Nature Genetics or published elsewhere. Should your manuscript be substantially delayed without notifying us in advance and your article is eventually published, the received date would be that of the revised, not the original, version.

Thank you for the opportunity to review your work.

Sincerely,

Michael Fletcher, PhD
Senior Editor, Nature Genetics

ORCID: 0000-0003-1589-7087

Referee expertise:

Referees #1, #3: telomere biology; human genetics, including GWAS.

Referee #2: telomere biology and genetics.

Reviewers' Comments:

Reviewer #1:

Remarks to the Author:

Notes on NG telomere paper review

In this manuscript, the authors report genetic analyses of telomere length measurements in the UK Biobank. The core novel aspect of this paper is the use of TelSeq, in conjunction with previously investigated qPCR measurements, to identify novel common, rare variant, and gene burden germline associations with telomere length. The genetic association methods appear sound and the use of data reduction approaches (PC1 and PC2) are an appropriate way of trying to classify shared and individual variability in the two main measures of telomere length.

Surprisingly, the authors find a low correlation (0.16) between the qPCR and TelSeq methods. This correlation is substantially lower than what has been observed in prior published studies that compare various methods of measuring telomere length. In fact, this correlation is so low that the authors refer to the methods as “orthogonal.” This seems like the wrong word choice for two reasons: there is a significant positive correlation (it’s not zero) and the methods are ostensibly trying to measure the same phenomenon.

Moreover, this disagreement between methods is a central problem that should be clarified and resolved prior to publication of the genetic associations identified in this manuscript. The low correlation raises concerns that the effect of experimental and other factors has not adequately been examined and adjusted for in the upstream analyses. There is limited description of the QC of the telomere length measurements in comparison to other methods employed in this paper and in comparison to similar efforts in other cohorts, such as the TopMed TelSeq paper (<https://doi.org/10.1016/j.xgen.2021.100084>), which also make it difficult to interpret the source of the discordance between the methods.

To assure readers of the legitimacy of the downstream findings in this paper, the authors should conduct a comprehensive series of QC checks and analyses to confirm (or change) the estimate of a low correlation between methods and to generate the most accurate phenotype possible for TelSeq TL. I propose approaches to do this below, but also encourage the authors to consider best approaches to support the robustness of the data underlying their findings.

1. Construct a flow chart of QC of TL measurements
2. Expand the methods section on the QC process for TL measurements—see the TopMed paper referenced above for minimal levels of description needed
3. Generate Bland-Altman plots comparing qPCR and TelSeq methods across the full sample and stratified by age, sex, and ancestry groups. The goal of this is to determine whether the variation between the measures is driven by individual strata or variables, which can inform the QC effort.
4. Evaluate outliers in each distribution and whether they are associated with any of the above listed stratification variables. This includes reporting on which individual measurements were removed and why—for example, it appears that there were more individuals missing TL data in the qPCR set than the TelSeq set—are these problematic samples or simply problematic (missing) measurements in one set?
5. Estimate the correlation of the two traits at different levels of adjustment, starting at the most raw data level and leading to the fullest model after adjustment for age, age², sex, BMI, ancestry, PCs, and any other factors that might be relevant. Does that correlation improve after taking into account these other factors?
6. Compare the distribution of PC1 and PC2 to the mean of TelSeq and qPCR and difference between TelSeq and qPCR. Presumably PC1 correlates strongly with the former and PC2 correlates with the latter.
7. Exclude all individuals with blood cancers from the dataset and re-run the core genetic association analyses. Check the NOTCH1 association here—it’s possible that some somatic signal is leaking through.
8. Same goes for CHIP—examine the distributions of TL in the full set versus a set that excludes any CHIP mutation carriers.

Minor comments:

409-410: “These individuals were pseudo randomly selected from the set of UKB participants.” This seems like text that applied to an earlier release of the WGS data rather than the full dataset used

here.

Figure 1A: The slope of the line doesn't appear to reflect the distribution, which may be obscured by the number of points. A contour plot that indicates density of points could make this clearer.

Reviewer #2:

Remarks to the Author:

The manuscript by Petrovski et al. follows on a large and existing body of literature to analyze genetic determinants of telomere length (TL) in UKB. The study combines data from whole genome sequencing and T/S PCR measurements of TL to infer genetic determinants. The combined approach expands the identifiable heritability of TL to ~36%. While the approach combining qPCR and TelSeq is interesting, I have several comments on the framework, context and novelty the study:

1. Nearly all the hits were identified in prior studies of either TL or clonal hematopoiesis (Codd 2021 and Kessler 2022) in the same UKB population. I am also not sure that the combined statistical approach really measures independent traits (this should be corrected in the text). But both have limitations: qPCR has a known higher error rate at the longer ranges of TL. More importantly, I am concerned about the novelty of the study especially since the authors rely heavily on the Mendelian context to verify findings and many of these concepts have been already synthesized in the Mendelian literature (point 4 below).
2. The data claiming an association between long TL and SRSF2 CHIP are outlier and warrant functional studies with mechanistic insight, especially in light of the discordance with the lower VAF trends in Figure 3B.
3. Telomere length is inherited independent of Mendelian traits. This is a well-established finding. The authors should mention this and related references for context, as not all TL is going to be related to sequence variants.
4. The citations included are incomplete. The first study to show association between CHIP and TL is not included (PMC7944936). Reviews cited on page 3, line 63 are outdated and refer to Mendelian randomization studies not established Mendelian patterns. The bidirectionality of rare variants in telomere genes including ACD from the Mendelian literature is well-known and has been synthesized and this reference inclusive of those citations should be added PMC10111244. The reference to short telomere-associated clonal mutations is also not the original study.

Reviewer #3:

Remarks to the Author:

Burren, Dhindsa, and Deevi, et al. report a large association study of leukocyte telomere length in a samples of nearly 500k individuals. Association analyses include common variants (GWAS approach), putatively pathogenic/functional rare variant testing (made feasible by the large sample size), and gene-level analysis based on collapsing ultra-rare/unique variants across subjects. They identify 162 common LTL-associated variants in GWAS, including 39 not previously implicated in telomere length regulation, as well as 4 additional novel loci in meta-analysis across racial/ethnic groups.

1. There have been many GWAS of common variants associated with LTL, including a recent one from this group and based on overlapping data which has identified the majority of the variants produced by this new analysis. While there are many novel items in this submission, I want to consider the

value of identifying new GWAS hits for LTL. One of the places that LTL GWAS hits have been most frequently utilized by the research community is in Mendelian randomization and polygenic risk score approaches for chronic diseases/cancer. The data in Supp Table 5 will assuredly be used for this purpose. As a minor issue, please note in the table how A0 and A1 are defined, and make clear that the Beta is relative to each additional copy of (ostensibly) A1.

2. In truth, the data in Supp T6 would be more powerful for PRS models based on its inclusion of more variants and their independent associations conditioned on additional signals in the region. Unfortunately, A0 and A1 data are missing from this table and should be provided.

3. There is a supp figure showing the p-values from the EUR vs. the multi-ethnic analyses, but this isn't super meaningful because the p-values are largely driven by the EUR subgroup. It would be helpful to also provide a similar figure, but for beta values in EUR vs. multi-ethnic.

4. The data in Table S5 include only results from the EUR analysis. Given historical difficulties in applying polygenic scores derived in European-ancestry populations to individuals of other racial/ethnic backgrounds, it would be very useful to show how well a PRS based on the multi-ethnic GWAS results does at discriminating inter-individual telomere length in each racial/ethnic subgroup in terms of variance explained. Other investigators can then use these data to study the effect of LTL-associated variants on other diseases/traits in non-white populations, accounting for potential limitations in model performance across ancestral background.

5. Finally, it would be most helpful to add columns to Table s5 that provide the Beta, Standard Error, and P-value in the AFR, AMR, ASJ, EAS, and SAS subjects at each of the sentinel variants discovered in EUR subjects.

The authors further identify 46 rare variants associated with LTL (located in 17 genes), including a novel association in NOTCH1. Thirty-two rare variants were associated with longer LTL, mostly in components of the shelterin and CST complex. Variants associated with shorter LTL were primarily located in genes previously associated with dyskeratosis congenita, a Mendelian disorder of telomere attrition. Additional insights come from distinct mutations in the same gene (eg, ACD), some of which were associated with longer LTL and others with shortened LTL that correspond to their functionally relevant locations in the protein domains that interact with different components of the shelterin complex. Importantly, all rare-variant associations were carefully controlled for potential linkage with common variants from the GWAS.

6. Although these are rare variants, in a sample this large I wonder if there was any evidence of homozygosity for a specific rare variant, or compound heterozygosity for two rare variants in the same gene (admittedly, they may be in-cis or in-trans and this won't be resolvable). If homozygosity was absent, did this represent a meaningful departure from Hardy-Weinberg equilibrium that could imply lethality?

Variant-collapsing analyses identified 18 genes associated with LTL, mostly in genes already identified in the rare-variant analyses. Two novel gene-level associations included G3BP1 and ZNF451. Because telomere length has been associated with clonal hematopoiesis, which could confound rare-variant associations, these associations were intersected with a list of known clonal hematopoiesis genes and

myeloid malignancy genes and re-evaluated using higher coverage WES data. Somatic variant calling confirmed that a subset of rare variant associations were driven by somatic variants underlying clonal hematopoiesis, which created associations with reduced LTL. However, DNMT3A somatic mutations were associated with longer LTL. In analysis of clone size, small clones were associated with longer LTL, suggesting that longer TL promotes acquisition of clonal hematopoiesis. However, small clones were associated with shorter TL in a subset of genes, indicating that certain CH subtypes can be promoted by shortened telomeres.

7. This is a well-conducted analysis and contains important insights for the field. While the list of candidate genes involved in CH and myeloid malignancy seems robust, was any consideration given to genes involved in lymphoid malignancy? While B and T cells represent a substantially smaller proportion of total blood-derived DNA, the authors are powered to detect very small effect sizes and the contributions of lymphoid malignancy genes should also be considered. Many of these genes that are relevant to CLL for instance, are already present in the list (eg, SF3B1, NRAS, TP53), but others are not. I'm most concerned about POT1, which is mutated in CLL, and about NOTCH1, which is both mutated in CLL and was identified as a novel LTL-associated gene in the rare variant analyses. Are the authors confident that the NOTCH1 association in rare variant analysis is not actually driven by a somatic alteration???

Finally, I really appreciated the authors approach to combining qPCR-based and TelSeq-based assessments of LTL and using PC1 as the primary outcome of interest in association tests. However, it does raise a couple of questions.

8. First, can the authors clarify if the a) the same DNA specimen was used for both the qPCR and WGS assays? If not the same specimen, was it from blood collected at an identical (ie, single) timepoint? If not, then associations between LTL and age should be clarified to reflect that this was accounted for in modeling.

9. I'd like to think more about how the qPCR and TelSeq measurements might meaningfully differ. There are at least two approaches to consider:

9a. Please identify the genes that were associated with qPCR-based LTL but not TelSeq-based LTL (collapsed from both common and rare variant approaches) and perform some sort of gene-set enrichment analysis. Then do the same for the complementary set of genes associated with TelSeq-based LTL but not qPCR-based LTL. If the genes are just generically associated with telomere pathways, then so be it. But it's possible that they may instead be associated with more informative pathways, like DNA repair mechanisms or cell-cycle control, which would help to inform on why the correlation between the two assays was quite modest.

9b. Have the authors considered how somatic loss of the X (or Y) chromosome might influence their results? Individuals with extensive somatic loss of X would have less total telomeric DNA, but the magnitude of this reduction should be different in TelSeq-based analyses than qPCR-based analyses. This is because the numerator – telomeric DNA – gets similarly reduced in both, but the denominator stays the same in qPCR (housekeeping gene) while the denominator is also reduced in TelSeq (total sequencing coverage). A number of GWAS hits for somatic loss of sex-chromosomes (see here: <https://pubmed.ncbi.nlm.nih.gov/28346444/>) overlap with your LTL loci. Would be worth seeing what

the p-values are for their 19 sentinel SNPs in your LTL GWAS, and look for heterogeneity in associations with qPCR vs TelSeq-based associations.

Author Rebuttal to Initial comments

Reviewer #1:

In this manuscript, the authors report genetic analyses of telomere length measurements in the UK Biobank. The core novel aspect of this paper is the use of TelSeq, in conjunction with previously investigated qPCR measurements, to identify novel common, rare variant, and gene burden germline associations with telomere length. The genetic association methods appear sound and the use of data reduction approaches (PC1 and PC2) are an appropriate way of trying to classify shared and individual variability in the two main measures of telomere length.

Surprisingly, the authors find a low correlation (0.16) between the qPCR and TelSeq methods. This correlation is substantially lower than what has been observed in prior published studies that compare various methods of measuring telomere length. In fact, this correlation is so low that the authors refer to the methods as “orthogonal.” This seems like the wrong word choice for two reasons: there is a significant positive correlation (it’s not zero) and the methods are ostensibly trying to measure the same phenomenon.

We have updated the referenced sentences accordingly:

*“As an alternative method for estimating TL, we also used TelSeq (**Supplementary Fig. 2**), which estimates telomere length from the whole-genome sequencing (WGS) data¹” (Page 4 line 20)*

and

“In a joint model, the association between each of the metrics and age remained highly significant, suggesting that each captures additional information.” (Page 5 line 7)

Moreover, this disagreement between methods is a central problem that should be clarified and resolved prior to publication of the genetic associations identified in this manuscript. The low correlation raises concerns that the effect of experimental and other factors has not adequately been examined and adjusted for in the upstream analyses. There is limited description of the QC of the telomere length measurements in comparison to other methods employed in this paper and in comparison to similar efforts in other cohorts, such as the TopMed TelSeq paper (<https://doi.org/10.1016/j.xgen.2021.100084>), which also make it difficult to interpret the source of the discordance between the methods.

To assure readers of the legitimacy of the downstream findings in this paper, the authors should

conduct a comprehensive series of QC checks and analyses to confirm (or change) the estimate of a low correlation between methods and to generate the most accurate phenotype possible for TelSeq TL. I propose approaches to do this below, but also encourage the authors to consider best approaches to support the robustness of the data underlying their findings.

1. Construct a flow chart of QC of TL measurements

We have constructed a flowchart detailing all QC and sample counts at each step as a new Supplementary Figure 1 shown below.

2. Expand the methods section on the QC process for TL measurements—see the TopMed paper referenced above for minimal levels of description needed

We now include a more thorough description of the QC throughout the methods section in addition to the additional QC chart per the above comment. We have also implemented a computationally intensive modified coverage correction scheme for TelSeq TL measurements as described in ² to mitigate any technical effects. This approach is described in the methods (Page 24 line 17):

“Correcting TelSeq TL estimates for technical confounders

We examined the correlation between inverse normal rank transformed TelSeq and UKB adjusted qPCR T/S (UKB showcase field id 22191) telomere length estimates finding modest agreement ($r^2=0.16$), perhaps indicating the presence of technical confounders.

*To mitigate this, for TelSeq TL estimates that were derived from WGS, we adapted the coverage correction method described by Taub et al.². Briefly, we used available Mosdepth³ coverage files available across 482,839 WGS samples, which given the scale were calculated using a `quantised` strategy, that merges adjacent bases if they fall in the same coverage bin. Overall, 4 read depth bins were selected ([0-9], [10-19], [20-49], [50+]). To compute overall coverage, we assumed that the coverage for a given base was the median of the read depth for that bin. As described in Taub et al. we split the genome (GRCh38) into 1Kb tiles and removed those that overlapped regions with poor mappability, that were blacklisted or overlapped known structural variants or were non-autosomal resulting in 178,120 1Kb bins (approx. 6% of the genome). Then for each sample we computed the average coverage across each bin. To facilitate downstream computation given the large size of the coverage matrix (i.e. 482,839 x 178,120), we investigated the performance of randomly batching samples for coverage adjustment (**Supplementary Note 2**). This supported a strategy of 24 randomised batches (23 batches of 20,000 and one batch of 22,839 participants). For each batch we used a randomised PCA approach implemented in the R package `rsvd`⁴ to estimate the first 300 principal components for each batch. To correct TelSeq TL estimates we then fit a linear model ($\text{TelSeq}_{\text{raw}} \sim \text{PC1-300}$) taking forward the resulting residuals as coverage corrected TelSeq TL estimates.”*

All analyses in the manuscript have been updated to use this coverage corrected TelSeq TL estimate.

3. Generate Bland-Altman plots comparing qPCR and TelSeq methods across the full sample and

stratified by age, sex, and ancestry groups. The goal of this is to determine whether the variation between the measures is driven by individual strata or variables, which can inform the QC effort.

Because the two metrics use different units (T/S ratio vs. basepairs), we used the inverse rank transform for both. An overall Bland-Altman plot of transformed and coverage adjusted TelSeq and qPCR metrics did not flag any obvious bias (0.0008 SD) or an obvious relationship between mean TL and TL difference between the two measures (Supplementary Figures 4-7 reproduced below)

Supplementary Figure 4: Bland-Altman plots comparing overall coverage adjusted TelSeq and qPCR TL estimates. To facilitate comparison at the same scale both metrics were inverse normal rank transformed before plotting. The black dashed lines from top to bottom represent 1.96 SD, mean and -1.96 SD respectively. Green, blue, and red shading indicates 95% confidence intervals associated with each of these. The bias (the difference between the expected difference (i.e. 0) and the observed mean) is shown in the top right corner. A positive value on the y-axis indicates a longer qPCR TL estimate for a participant than coverage corrected TelSeq TL metric.

Supplementary Figure 5: Bland-Altman plots comparing coverage adjusted TelSeq and qPCR TL estimates by sex. The legends are the as for Supplementary Figure 4.

Supplementary Figure 6: Bland-Altman plots comparing coverage adjusted TelSeq and qPCR TL estimates by ancestry. The legends are the as for Supplementary figure 4.

Supplementary Figure 7: Bland-Altman plots comparing coverage adjusted TelSeq and qPCR TL estimates by age strata. The legends are the as for Supplementary figure 4.

We have added these figures and a description of their interpretation to the methods of the main manuscript (Page 25 Line 17):

*“To assess coverage corrected TelSeq TL estimates we created Bland-Altman plots stratified by sex, ancestry and age using inverse normal transformed metrics for 462,666 participants where both metrics were available (**Supplementary Fig. 4-7**). Overall, we observed little bias in TL estimates when comparing qPCR and TelSeq methods.”*

4. Evaluate outliers in each distribution and whether they are associated with any of the above listed stratification variables. This includes reporting on which individual measurements were removed and why—for example, it appears that there were more individuals missing TL data in the qPCR set than the TelSeq set—are these problematic samples or simply problematic (missing) measurements in one set?

We used logistic regression to evaluate overall outliers identified through our BA plots and their association with the different strata above (Table 1).

Table 1: Association between age, sex and ancestry and BA plot outliers. Ancestry Odds Ratios are with respect to NFE ancestry individuals.

Strata	Odds Ratio	P
Sex (Baseline Male)	1.01	0.7
Age	0.98	0.03
Ancestry AMR	0.83	0.31
Ancestry ASJ	0.96	0.73
Ancestry EAS	1.21	0.03
Ancestry SAS	0.95	0.25
Ancestry AFR	1.80	1.16 x 10⁻¹²

From this analysis, we detected only one significant association after Bonferroni correction, which was between AFR ancestry strata and outlier status (Table 1). This might be expected given the strong associations between common and rare variants in the HBB locus and telomere length measured by qPCR, which is not found when we use TelSeq WGS as a substrate, as it does not rely on this gene for standardisation. To test this, we reran this analysis with an additional covariate as to whether an individual was a carrier of a rare HBB variants across all ancestries. We found that this significantly attenuated the association ($OR_{AFR} 1.23$, $P_{AFR}=2.65 \times 10^{-5}$) between outlier and AFR status, indicating that the observed bias for qPCR estimates for the AFR strata might be explained by this technical effect.

We have added to the methods the following text to describe this (page 25 line 20):

“We used logistic regression to assess whether outlier status (by difference) was significantly associated with any of these biological metrics. Only AFR genetic ancestry was significantly associated with outlier status ($P=1.16 \times 10^{-12}$ OR = 1.80), however when we added rare HBB coding variant carrier status into the model this association was significantly attenuated ($P=2.65 \times 10^{-5}$ OR = 1.23) indicating this might be driven by the genetic effects reported for qPCR TL within the HBB locus.”

In reference to the reviewer’s comment about the missing samples data for the qPCR set. Our main reason for excluding was because the PCA approach we adopted to combine qPCR and TelSeq measurements does not handle missing data. However, we have added the following section to the methods (page: 24 line 7):

“qPCR TL estimates

For qPCR TL measurements we used those available through UKB (Field ID 22191) derived from baseline samples for a total of 472,518 participants. We used rank inverse normal transformed T/S adjusted ratios without further adjustment given the extensive QC already performed on these measurements⁵. In total we identified 9,852 (2%) of the qPCR samples that lacked a matching TelSeq TL estimate from downstream analyses. We found that qPCR measurements in this set were significantly longer when compared to samples with both TelSeq and qPCR metrics (t-test $P=8.86 \times 10^{-42}$, $mean_{TelSeq\&qPCR}=5.0 \times 10^{-4}$, $mean_{qPCR,only}=0.14$).

As described above, we did find a significantly higher TL measurement for those samples where only a qPCR measurement was available, justifying their removal. A reciprocal comparison of samples with a TelSeq but no qPCR TL measurement did not show significant differences (t-test $P=0.17$, $mean_{TelSeq\&qPCR}=5.0 \times 10^{-4}$, $mean_{TelSeq,only}=7.0 \times 10^{-3}$).

5. Estimate the correlation of the two traits at different levels of adjustment, starting at the most raw data level and leading to the fullest model after adjustment for age, age², sex, BMI, ancestry, PCs, and any other factors that might be relevant. Does that correlation improve after taking into account these other factors?

We followed the reviewer’s recommendation as detailed above. Overall, none of these models improved the correlation between qPCR and TelSeq TL estimates. We thus undertook an alternative approach, where we first examined several relevant variables (including those suggested by the reviewer) as to whether they might improve correlation on their own (Table 2). We Found that four technical variables (Total reads, Sequencing processing pipeline, Coverage

Uniformity and Sequencing Provider) did marginally improve the correlation, so we took these forwards for variable selection.

Table 2 Correlation between qPCR and TelSeq TL estimates after adjusting for covariates.

Model Term	r^2 (qPCR vs TelSeq)	Change in r^2
No covariate	0.2882	
wgs.total.reads	0.2883	0.0001
coverage.uniformity	0.2882	0.0000
SequencingProvider	0.2882	0.0000
Sequencing Pipeline	0.2882	0.0000
Ancestry PC[1:4]	0.2875	-0.0007
BMI	0.2859	-0.0023
Sex	0.2836	-0.0046
Age	0.2543	-0.0339
Age ²	0.2543	-0.0339
Age*Sex	0.2502	-0.0380
Age ² *Sex	0.2502	-0.0380

To augment the above we used simple stepwise regression as implemented in the R 'MASS' package to perform variable selection, and the final set of variables included Sequencing Provider, Total read count and coverage uniformity as used in the original submission. These were all used in the first submission, and so the maximum correlation achieved was the same ($r^2 = 0.288$).

We also took the opportunity to assess 19 WGS metrics captured by our sequencing pipeline to see whether there were additional technical confounders that we might have missed. This is described in the methods (page 27 line 8):

“Finally, we looked for univariate association between 19 WGS sequence metrics (Supplementary Table 3) collected on each sample and both qPCR and coverage

corrected TL estimates, using scaled WGS sequence metrics and inverse normal rank transformed qPCR and coverage adjusted TL estimates to facilitate comparison.

*We found that that overall, 14 and 16 WGS metrics were significantly associated with coverage adjusted TelSeq and qPCR TL measurements (**Supplementary Fig 8, Supplementary Table 3**) respectively. Of these many were highly correlated, however we noted that a combination of coverage uniformity, total WGS reads, and sequencing pipeline captured these, and so were included as covariates in downstream analyses (**Supplementary Note 3**)."*

These analyses and a description of the technical measures considered are available in Supplementary Table 3, Supplementary Figure 8 and Supplementary Note 3 reproduced below:

"Supplementary Note 3: on adjusting WGS TelSeq estimates for technical confounders.

*Given the relatively low correlation between qPCR and WGS TelSeq TL estimates we sought to understand whether this could be due to various technical confounders. We captured 19 sequencing covariates for all 462,666 samples (**Supplementary Table 1**) where TL measurements were available for TelSeq and qPCR methods. Initially examined these variables in a univariate framework to understand which of these variables were significantly associated with either raw TelSeq, adjusted qPCR, or coverage adjusted TelSeq TL metrics. To do this we inverse normal transformed each TL outcome variables and then regressed each technical covariate in turn resulting 19 x 3 univariate associations (TelSeq, qPCR and PC1). From these we selected those technical variables that were Bonferroni significant (**Supplementary Table 3**). For comparison we also added age and sex as these have been previously associated in multiple published studies with TL. Overall (excluding age and sex) we detected 20 technical variables that on their own were significantly associated with at least one of the TL metrics (**Supplementary Fig. 8**). To understand the suitability of adjusting for these in our downstream analysis we performed three analyses. Firstly, we assessed the correlation between qPCR and coverage adjusted TelSeq TL without controlling for any of the technical variables identified above, using this as a baseline ($r^2=0.288$). Using a linear model we next regressed out either all the significant technical variables or selected technical variables that were not themselves correlated with age or sex and repeated the correlation analysis. We found that adjusting for these technical confounders made little difference to qPCR correlation (all variables $r^2=0.273$), (selected variables $r^2=0.288$). For this reason, we decided to use just the baseline coverage TelSeq WGS in downstream analysis."*

6. Compare the distribution of PC1 and PC2 to the mean of TelSeq and qPCR and difference between TelSeq and qPCR. Presumably PC1 correlates strongly with the former and PC2 correlates with the latter.

Supplementary Figure 9: Comparison of PC1 and PC2 rotations. Mean (Top) and difference between qPCR and TelSeq TL (Bottom) metrics are plotted. Dotted lines indicate $x=y$ (Top) and $x=-y$ (Bottom) and are included for reference.

Exactly, PC1 captures the mean TL across the two measures whilst minimising the correlation with differences, whereas PC2 does the reciprocal and captures the differences whilst minimising the correlation with mean. We have included this plot as Supplementary Figure 9 (reproduced above) to facilitate interpretation of the different PCs for readers. To link this to the main text we have added the following sentence (page 27 line 19):

“Overall PC1 was highly correlated with the standardized mean across TelSeq and qPCR metrics whereas PC2 was correlated with their difference (Supplementary Fig. 9).”

7. Exclude all individuals with blood cancers from the dataset and re-run the core genetic association analyses. Check the NOTCH1 association here—it’s possible that some somatic signal is leaking through.

We thank the reviewer for this suggestion. Although in the manuscript we attempted to build a VAF classifier to exclude associations where there was obvious VAF skew, this approach is less robust to very rare variants, such as the one observed for NOTCH1 (6 carriers). In the updated manuscript, we excluded all individuals with a haematological malignancy diagnosis from all rare variant analyses per the reviewer’s suggestion. The NOTCH1 association was no longer present, indicating that this was driven by somatic signal. We have removed all references to this in the manuscript. We have made this clear in the main text (page 8 line 3):

“After removing individuals with known haematological malignancies at sampling (N=3,073), we performed both variant-level (exome wide association study, ExWAS) and gene-level (rare variant aggregated collapsing analyses) as previously described⁶.”

8. Same goes for CHIP—examine the distributions of TL in the full set versus a set that excludes any CHIP mutation carriers

We reran ExWAS and Collapsing genetic analyses excluding all 12,582 NFE genetic ancestry individuals for which we detected CH. Overall, we found that all associations reported in the manuscript were robust, with the expectation that we expect a slight attenuation in p-values concomitant with a reduction in power due to the lower sample size (Figure 1).

Figure 1: Comparison of rare variant analyses with and excluding Haematological malignancies

Minor comments:

409-410: "These individuals were pseudo randomly selected from the set of UKB participants." This seems like text that applied to an earlier release of the WGS data rather than the full dataset used here.

This was an error referring to an earlier draft and we have removed this sentence accordingly.

Figure 1A: The slope of the line doesn't appear to reflect the distribution, which may be obscured by the number of points. A contour plot that indicates density of points could make this clearer.

We thank the reviewer for this suggestion we have replaced the plot in Fig 1A (adapted version shown below in Figure 2) with a Hexagonal heatmap of 2d bin counts. With regards to slope of the line not appearing to fit the distribution, we have doubled checked this and it is indeed the true fit from OLS/ linear modelling (beta=0.54, $R^2=0.29$) and Pearson correlation. For comparison we also plot the singular value decomposition (SVD) fit in black, this is the result of performing SVD on the variance/covariance matrix of qPCR and TelSeq TL values, then selecting the dominant vector from the resultant right singular matrix (V) and using this compute a slope. The SVD fit appears visually to better represent the relationship between the two metrics perhaps because it less susceptible to leverage of points outside of the main distribution. To prevent confusion, we have stuck to using the linear model fit in the main figure given to agree with the R^2 value annotated on the plot.

Figure 2 Correlation between transformed qPCR and WGS Telseq telomere length metrics. The orange and black dashed lines indicate ordinary least squares and SVD lines of best fit respectively.

Reviewer #2:

The manuscript by Petrovski et al. follows on a large and existing body of literature to analyze genetic determinants of telomere length (TL) in UKB. The study combines data from whole genome sequencing and T/S PCR measurements of TL to infer genetic determinants. The combined approach expands the identifiable heritability of TL to ~36%. While approach combining qPCR and TelSeq is interesting, I have several comments on the framework, context and novelty the study:

1. Nearly all the hits were identified in prior studies of either TL or clonal hematopoiesis (Codd 2021 and Kessler 2022) in the same UKB population. I am also not sure that the combined statistical approach really measures independent traits (this should be corrected in the text). But both have limitations: qPCR has a known higher error rate at the longer ranges of TL. More importantly, I am concerned about the novelty of the study especially since the authors rely heavily on the Mendelian context to verify findings and many of these concepts have been already synthesized in the Mendelian literature (point 4 below).

It is important to observe many of the associated loci having been described in prior studies, which offer good positive controls. In the updated manuscript, we better emphasize novel signals. We emphasize three novel signals from the rare variant collapsing analysis, including G3BP1, BRIP1, and ZNF451, highlighting the critical role of stress/DNA damage response genes in regulating TL. In addition to the novel observations between CH and TL, we have now integrated plasma proteomic data to shed light on 9 proteins whose abundance has a causal relationship for TL. These include existing therapeutic targets (e.g. PARP1), proteins associated with nucleotide metabolism (CDA, TK1), and other novel causal relationships that may be worthy of future consideration for therapeutic targeting. We describe this in more detail in the main text (page 14 line 1):

“Identification of causal relationships between plasma proteome and telomere length.

We integrated pQTL data from the UKB PPP project that examined genetic associations between approximately 3,000 plasma proteins⁷ with our TL PC1 genetic associations. Across all PC1 GWAS significant loci we identified 2,905 overlapping pQTL loci ($p < 1.7 \times 10^{-11}$) (Supplementary Table 11). We used coloc⁸ to assess each of these and found strong evidence for a shared causal variant modulating both TL and plasma protein abundance at 266 (9%) of these overlaps. Of these, ten were colocalizations in cis and 256 were in trans. For the cis signals we used pQTLs as instruments in a Mendelian randomisation analysis (Methods) to assess whether plasma proteome abundance might be causally related to telomere length. We found evidence for a causal relationship between nine protein assays and TL after multiple testing correction (Supplementary Table 12, Supplementary Fig. 19), including some well-established telomere-related proteins (e.g., TK1, CDA and PARP1). For TK1, SPRED2, and BCL2L15, MR-Egger analysis highlighted the potential presence of pleiotropy, which might invalidate MR

assumptions. One protein, RPA2, binds single stranded DNA to protect from instability and breakage, and recently has been shown to be involved in telomere maintenance⁹. The remaining associations appear novel and warrant future functional studies to elucidate the mechanism by which they mediate telomere length. Of the trans colocalising proteins, 183/256 (71%) were found in the 12q24.12 locus containing SH2B3, which is known to be highly pleiotropic. Of the remaining trans colocalising protein assay associations, six exhibited colocalisation with more than one locus (Supplementary Fig. 20). These included FLT3LG for which trans pQTL signals colocalised with variants in ATM, TERT, and SETBP1 loci.

We also examined overlap between our rare variant TL analyses and rare pQTLs described in Dhindsa et al.¹⁰. At the variant level, no germline overlapping variants were identified. At the gene-level we identified one significant and two suggestive overlapping signals between a pQTL and PC1 TL. The significant association implicated a trans association between rare loss of function variants in TERT associated with shorter TL (ptvraredmg $P=1.7 \times 10^{-134}$ beta=-0.52 [-0.70 to -0.59]) and increased FLT3LG plasma abundance (ptvraredmg $P=4.68 \times 10^{-9}$ beta=0.52 [0.35 to 0.69]). The remaining suggestive associations overlapped with trans pQTLs for Alpha-fetoprotein abundance (AFP), with putative loss of function for ATM and ZNF451 being associated with shorter TL (ptvraredmg $P=5 \times 10^{-27}$ beta=-0.15 [-0.18 to -0.12]) and increased AFP (ptvraredmg $P=1.2 \times 10^{-8}$ 0.25[0.16 to 0.34]) and longer TL (ptv $P=1.1 \times 10^{-11}$ 0.36 [0.25 to 0.46]) and decreased AFP (ptv $P=4.9 \times 10^{-7}$ -0.76 [-1.1 to -0.47]) respectively.”

In response to editorial suggestions we have also integrated various functional data sources with GWAS summary statistics in order to prioritise genes and pathways, this is described in the manuscript as follows (page 15, line 15):

“Causal gene prioritisation

To prioritise putative causal genes in GWAS loci we generated a list of 7,334 protein coding genes overlapping a TL PC1 locus and annotated this gene set with data integrated from seven separate sources (**Methods**). Assuming equal weighting across all seven prioritisation categories we computed a simple sum to prioritise genes within each PC1 GWAS locus. Of the 7,334 protein coding genes considered 404 had a prioritisation score > 0 and a single gene was prioritised in 94 of the 192 PC1 TL GWAS loci (**Supplementary Table 14**). We found that these prioritised genes were more enriched ($p=4.12 \times 10^{-15}$) for the Reactome pathway “Extension of Telomeres” (R-HAS-180786) compared to 50 gene sets of the same size derived from randomly sampled

closest genes ($p_{\text{median}} = 5.6 \times 10^{-5}$) (**Supplementary Fig. 21 and Supplementary Table 15**).”

Overall, we believe this novel TelSeq TL estimate will have utility to the field in general, and we will make this available to global researchers via the UKB returned data mechanism. Additionally full GWAS summary statistics for TL PC1 and TL PC2 will be made available via the GWAS catalogue, and rare variant associations ($p < 1e-4$) will be made available via the AZ Portal (azphewas.com).

2. The data claiming an association between long TL and SRSF2 CHIP are outlier and warrant functional studies with mechanistic insight, especially in light of the discordance with the lower VAF trends in Figure 3B.

We agree that functional studies are critical for mechanistic insight of this association. While functional studies are not in the scope of this current manuscript, we highlight the need for future studies to validate the SRSF2 association and to elucidate other TL mechanisms more broadly in the revised discussion (Page 21 line 14).

“The possibility that SRSF2 mutations confer advantage through telomere modulation offers a novel explanation for the expansion of these mutant clones specifically in older age, however further functional studies will be required to validate and elucidate the underlying biological mechanisms involved.”

3. Telomere length is inherited independent of Mendelian traits. This is a well-established finding. The authors should mention this and related references for context, as not all TL is going to be related to sequence variants.

We agree that telomere length genetic architecture is polygenic in nature, as demonstrated by ours and other studies. It is also well established that there are many factors outside of sequence variants that modulate telomere length, and in our study we show that using common variants we are only able to explain approximately 10% of the trait heritability using our PC1 metric. To highlight that there are well-established non sequence variant contributors to telomere length, including epigenetic modifications and lifestyle factors like smoking and stress we have added the following to text to the manuscript (Page 3 line 7):

“Telomere length (TL) demonstrates considerable interindividual variability modulated by heritable^{11,12}, environmental and lifestyle factors such as smoking behaviour and stress¹³”

4. The citations included are incomplete. The first study to show association between CHIP and TL is not included (PMC7944936). Reviews cited on page 3, line 63 are outdated and refer to Mendelian randomization studies not established Mendelian patterns. The bidirectionality of rare variants in telomere genes including ACD from the Mendelian literature is well-known and has been synthesized and this reference inclusive of those citations should be added PMC10111244. The reference to short telomere-associated clonal mutations is also not the original study.

We thank the reviewer for bringing these unfortunate omissions to our attention. We have now rectified with complete citations. For the reference to short telomere-associated clonal mutations we have now cited (page 17 line 8): “Schratz,K.E. et al. (2020) Cancer spectrum and outcomes in the Mendelian short telomere syndromes. Blood, 135, 1946–1956.” and (page 18 line 6) “Gutierrez-Rodrigues,F. et al. (2021) Clonal hematopoiesis in telomere biology disorders associates with the underlying germline defect and somatic mutations in POT1, PPM1D, and TERT promoter. Blood, 138, 1111–1111.”

Reviewer #3:

Burren, Dhindsa, and Deevi, et al. report a large association study of leukocyte telomere length in a samples of nearly 500k individuals. Association analyses include common variants (GWAS approach), putatively pathogenic/functional rare variant testing (made feasible by the large sample size), and gene-level analysis based on collapsing ultra-rare/unique variants across subjects. They identify 162 common LTL-associated variants in GWAS, including 39 not previously implicated in telomere length regulation, as well as 4 additional novel loci in meta-analysis across racial/ethnic groups.

1. There have been many GWAS of common variants associated with LTL, including a recent one from this group and based on overlapping data which has identified the majority of the variants produced by this new analysis. While there are many novel items in this submission, I want to consider the value of identifying new GWAS hits for LTL. One of the places that LTL GWAS hits have been most frequently utilized by the research community is in Mendelian randomization and polygenic risk score approaches for chronic diseases/cancer. The data in Supp Table 5 will assuredly be used for this purpose. As a minor issue, please note in the table how A0 and A1 are defined, and make clear that the Beta is relative to each additional copy of (ostensibly) A1.

We have updated the supplementary table description to read:

“GWAS index variants from REGENIE analysis of PC1, PC2, qPCR and TelSeq TL metrics in NFE ancestry individuals. Effect size/beta is with respect to each additional copy of A1 allele”

2. In truth, the data in Supp T6 would be more powerful for PRS models based on its inclusion of more variants and their independent associations conditioned on additional signals in the region. Unfortunately, A0 and A1 data are missing from this table and should be provided.

We have now added in the relevant allele assignments to this table to facilitate the building of PRS models based on these data. We have updated the supplementary table description to read:

“Conditional analysis using GCTA COJO for PC1 and PC2 significant ($p < 5 \times 10^{-8}$) loci. Effect size/beta is with respect to each additional copy of A1 allele.”

3. There is a supp figure showing the p-values from the EUR vs. the multi-ethnic analyses, but this isn't super meaningful because the p-values are largely driven by the EUR subgroup. It would be helpful to also provide a similar figure, but for beta values in EUR vs. multi-ethnic.

We have added an additional panel into Supplementary Figure 22 with this information which we reproduce below:

Supplementary Figure 22: Comparison of p values for NFE and fixed effect cross ancestry meta-analysis collapsing analysis (NFE, AFR, SAS, EAS, and ASJ) for TL PC1. Only variants $p_{NFE} < 5 \times 10^{-5}$ are shown, panel A significance and panel B effect (TL SD).

4. The data in Table S5 include only results from the EUR analysis. Given historical difficulties in applying polygenic scores derived in European-ancestry populations to individuals of other racial/ethnic backgrounds, it would be very useful to show how well a PRS based on the multi-ethnic GWAS results does at discriminating inter-individual telomere length in each racial/ethnic subgroup in terms of variance explained. Other investigators can then use these data to study the effect of LTL-associated variants on other diseases/traits in non-white populations, accounting for potential limitations in model performance across ancestral background.

To examine this, do this we took the entire cohort, regardless of ancestry and randomly split it into non-overlapping training (80%) and test (20%) sets. Using the training set, we performed a GWAS using REGENIE using the same covariates as described previously in the manuscript for

the PC1 phenotype, with the addition of indicator variables to model over-arching ancestral groups. We took the summary statistics from this analysis as input to PRS-CS¹⁴ using the recommended 1KG EUR panel as the LD reference to construct a genome-wide PRS. We applied this PRS to compute a genetically predicted TL metric (based on PC1) for all test set samples. We computed the correlation of ‘measured’ PC1 TL with genetically predicted PC1 for each ancestry strata which is shown below (Table 3).

Table 3 Results of PRS prediction in different ancestries in a held-out test set.

Ancestry	r^2	r	p	Lower	upper	N_{Test}
NFE	0.06	0.24	0.00E+00	0.23	0.25	87,650
SAS	0.04	0.21	3.90E-20	0.16	0.25	1,907
AFR	0.02	0.16	3.38E-10	0.11	0.20	1,588
EAS	0.04	0.21	5.84E-06	0.12	0.29	472
AMR	0.10	0.31	2.31E-04	0.15	0.45	138
ASJ	0.03	0.17	5.99E-05	0.09	0.25	536

As expected, given the large imbalance in the training set towards NFE individuals coupled with using 1KG EUR genetic ancestry individuals in the LD reference set to train the PRS, the PRS explained the most variance in NFE strata. One potential outlier was for AMR ancestry participants, but we note that the number of individuals in the training set is modest and the point estimate for the Pearson correlation is accompanied by very large confidence interval. Overall, these results confirm the many reports in the literature¹⁵ that also describe how the performance of PRS's trained on one ancestry are attenuated when applied to another.

5. Finally, it would be most helpful to add columns to Table s5 that provide the Beta, Standard Error, and P-value in the AFR, AMR, ASJ, EAS, and SAS subjects at each of the sentinel variants discovered in EUR subjects.

We have implemented this as described by looking up NFE index variant in the various ancestry strata and adding these to Supplementary Table 5 as additional columns.

The authors further identify 46 rare variants associated with LTL (located in 17 genes), including a

novel association in NOTCH1. Thirty-two rare variants were associated with longer LTL, mostly in components of the shelterin and CST complex. Variants associated with shorter LTL were primarily located in genes previously associated with dyskeratosis congenita, a Mendelian disorder of telomere attrition. Additional insights come from distinct mutations in the same gene (eg, ACD), some of which were associated with longer LTL and others with shortened LTL that correspond to their functionally relevant locations in the protein domains that interact with different components of the shelterin complex. Importantly, all rare-variant associations were carefully controlled for potential linkage with common variants from the GWAS.

6. Although these are rare variants, in a sample this large I wonder if there was any evidence of homozygosity for a specific rare variant, or compound heterozygosity for two rare variants in the same gene (admittedly, they may be in-cis or in-trans and this won't be resolvable). If homozygosity was absent, did this represent a meaningful departure from Hardy-Weinberg equilibrium that could imply lethality?

We examined all the variants from the PC1 TL exWAS rare TL hits and found missense homozygotes/compound heterozygotes for individuals in ACD, DCLRE1B, JAK2, POT1, PPDPF, SAMHD1, TERF1, and TERT. No germline rare variants showed significant departure from HWE. To highlight this point, we have added two extra columns to Supplementary Table 7 that contain HWE P and whether a homozygous individual was identified for that variant.

Variant-collapsing analyses identified 18 genes associated with LTL, mostly in genes already identified in the rare-variant analyses. Two novel gene-level associations included G3BP1 and ZNF451. Because telomere length has been associated with clonal hematopoiesis, which could confound rare-variant associations, these associations were intersected with a list of known clonal hematopoiesis genes and myeloid malignancy genes and re-evaluated using higher coverage WES data. Somatic variant calling confirmed that a subset of rare variant associations were driven by somatic variants underlying clonal hematopoiesis, which created associations with reduced LTL. However, DNMT3A somatic mutations were associated with longer LTL. In analysis of clone size, small clones were associated with longer LTL, suggesting that longer TL promotes acquisition of clonal hematopoiesis. However, small clones were associated with shorter TL in a subset of genes, indicating that certain CH subtypes can be promoted by shortened telomeres.

7. This is a well-conducted analysis and contains important insights for the field. While the list of candidate genes involved in CH and myeloid malignancy seems robust, was any consideration given to genes involved in lymphoid malignancy? While B and T cells represent a substantially smaller proportion of total blood-derived DNA, the authors are powered to detect very small effect sizes and the contributions of lymphoid malignancy genes should also be considered. Many of these genes that are relevant to CLL for instance, are already present in the list (eg, SF3B1, NRAS, TP53), but others are

not. I'm most concerned about POT1, which is mutated in CLL, and about NOTCH1, which is both mutated in CLL and was identified as a novel LTL-associated gene in the rare variant analyses. Are the authors confident that the NOTCH1 association in rare variant analysis is not actually driven by a somatic alteration???

We thank Reviewer 3 for these insightful comments and questions about CH.

For this analysis, we focussed on CH related to myeloid expansions, where the defining driver genes and variant-calling methods are supported by an established and robust literature, as compared to lymphoid. Our primary objective was to uncover relationships between telomere length and driver-specific subtypes of CH, and a critical component of the discovery approach was stratifying by CH clone size. In the lymphoid lineage, this approach would be severely limited when using whole-blood-derived DNA sequencing; the contribution of lymphocytes to whole blood leukocytes is comparatively small (as Reviewer 3 points out), only allowing for levels of sensitivity suitable for detection of larger clones.

As pointed out, some genes in the 'myeloid' gene panel can also drive lymphoid malignancies (such as SF3B1, NRAS and TP53). Indeed, using bulk blood DNA sequencing, where there are shared drivers, it is essentially impossible to differentiate the cell lineage with any certainty, and this is an issue across most of the published literature on CH. However, we believe that the impact for our study is likely very small because: (i) small gene-specific variants only drive a minority of lymphoid expansions (while larger chromosomal aberrations are more common drivers), and (ii) our sensitivity to detect somatic mutations in the lymphoid lineage would have been limited (as discussed above).

It will be important in future studies to understand whether the driver-gene-specific observations we make in CH (myeloid) clonal expansions are also observed in cancers affecting other cell types / tissues that share the same driver genes, including lymphoid malignancies. This would imply fundamental impacts on stem cell/progenitor biology that operate across tissues, with broad scientific and potentially therapeutic implications. Specifically in relation to POT1 and NOTCH1, as previously discussed with Reviewer 1, we have redone all rare variant analyses excluding all individuals with a haematological malignancy. In this analysis NOTCH1 is no longer significantly associated indicating that the signal was most likely due to somatic mutation. ExWAS POT1 associations did not show evidence of significant VAF skew (Supplementary Fig. 15). In addition, there was no evidence for an association between POT1 variant carriers and age ($p=0.632$, $\beta=-0.122$, $\beta.se=0.26$). We therefore feel confident that POT1 associations are likely due to germline rather than somatic contamination.

Finally, I really appreciated the authors approach to combining qPCR-based and TelSeq-based assessments of LTL and using PC1 as the primary outcome of interest in association tests. However, it does raise a couple of questions.

8. First, can the authors clarify if the a) the same DNA specimen was used for both the qPCR and WGS assays? If not the same specimen, was it from blood collected at an identical (ie, single) timepoint? If not, then associations between LTL and age should be clarified to reflect that this was accounted for in modeling.

For samples with qPCR TL metrics, we selected only measurements taken from a baseline sample. We discussed with UKB but unfortunately information on whether WGS was performed on the same aliquot or one taken at the same time point is not available for WGS samples at this time. However, they did reassure us that the overwhelming majority of WGS were performed on baseline samples and therefore should match qPCR in terms of sampling time. Considering this we feel the potential for this limitation to undermine our analyses and conclusions is minimal, especially because of the joint TL metric approach we employed.

9. I'd like to think more about how the qPCR and TelSeq measurements might meaningfully differ. There are at least two approaches to consider:

9a. Please identify the genes that were associated with qPCR-based LTL but not TelSeq-based LTL (collapsed from both common and rare variant approaches) and perform some sort of gene-set enrichment analysis. Then do the same for the complementary set of genes associated with TelSeq-based LTL but not qPCR-based LTL. If the genes are just generically associated with telomere pathways, then so be it. But it's possible that they may instead be associated with more informative pathways, like DNA repair mechanisms or cell-cycle control, which would help to inform on why the correlation between the two assays was quite modest.

We thank the reviewer for this suggestion, which we have implemented as described. Of course, for common variant analyses often non-coding variants are discovered and ascribing the causal gene is non-trivial (as you can see for PC1 we have put some effort into applying more sophisticated techniques that integrate functional data to do this). For this analysis we took the simple approach of assigning the closest gene as putatively causal for GWAS, as there is some literature estimating that this approach can select the causal gene in approximately 2/3 of cases¹⁶. This led to 150 genes for qPCR and 199 genes for TelSeq after combining with collapsing and exWAS identified genes. For qPCR there were 53 genes not found in the TelSeq gene set and there were 102 genes found in TelSeq but not in the qPCR set. We took forward both exclusive sets for enrichment analysis in the Reactome pathway resource using the ReactomePA¹⁷ R library. At a Bonferroni adjusted p value <0.05 we found 17 pathways enriched for TelSeq and none for qPCR exclusive genes. The TelSeq-exclusive genes were enriched for key ontologies related to telomere biology, including cell cycle regulation, DNA replication and repair mechanisms, stress response pathways, transcriptional regulation, and oncogenic signalling processes.

9b. Have the authors considered how somatic loss of the X (or Y) chromosome might influence their results? Individuals with extensive somatic loss of X would have less total telomeric DNA, but the magnitude of this reduction should be different in TelSeq-based analyses than qPCR-based analyses. This is because the numerator – telomeric DNA – gets similarly reduced in both, but the denominator stays the same in qPCR (housekeeping gene) while the denominator is also reduced in TelSeq (total sequencing coverage). A number of GWAS hits for somatic loss of sex-chromosomes (see here: <https://pubmed.ncbi.nlm.nih.gov/28346444/>) overlap with your LTL loci. Would be worth seeing what the p-values are for their 19 sentinel SNPs in your LTL GWAS, and look for heterogeneity in associations with qPCR vs TelSeq-based associations.

We performed two analyses to investigate this idea as described in Supplementary Note 4 which we reproduce below:

“Supplementary Note 4: on the effect of mosaic loss of chromosome X and Y on qPCR and WGS TelSeq TL estimates.

Mosaic loss of X (mLoX) or Y (mLoY) chromosomes could theoretically lead to differences in TL estimates between qPCR and TelSeq approaches, which could underlie some of the differences we observed between these metrics. In both metrics, the numerators are a measure of telomere content. However, for qPCR-derived estimates, the denominator is a measure of the abundance of the single copy gene (S) HBB, which is expected to be unaffected by mLoX/Y. For TelSeq, the denominator reflects the total GC adjusted WGS read count and is expected to be attenuated in the event of significant mLoX/Y. Under these assumptions we might therefore expect shorter TL estimates from qPCR than TelSeq in the presence of mLoX/Y.

To test this, we considered 19 sentinel germline variants that were associated with LoY in a prior GWAS¹⁸. Of these 19, 11 were also significantly associated ($P_{Bonferroni} < 0.003$) with TL. Under the hypothesis that mLoY would bias estimates, we would expect to identify significantly different effect size estimates for qPCR versus TelSeq. However, there was no overall evidence of effect size heterogeneity between metrics (Table 1), suggesting that mLoY is not systematically biasing TL estimates.

Supplementary Note 4: Table 1 Results of Heterogeneity test of effect sizes between TelSeq and qPCR TL effect estimates for variants also associated with mLoY, effect allele is a1.

rs#	Variant (chr-pos-a0-a1)	Cochrane's Q	P heterogeneity	Adjusted P heterogeneity
rs13191948	6-109313396-T-C	4.56	0.02	0.18
rs4754301	11-108177814-A-G	3.17	0.04	0.41
rs13088318	3-101523907-G-A	2.89	0.04	0.49
rs381500	6-164057356-A-C	2.35	0.06	0.69
rs11082396	18-44500755-C-T	2.07	0.08	0.83
rs1122138	14-95713905-A-C	0.32	0.28	1.00
rs17758695	18-63253621-T-C	0.82	0.18	1.00
rs2736609	1-156232849-T-C	0.29	0.30	1.00
rs77522818	17-49740011-T-A	1.62	0.10	1.00
rs10687116	13-41103945-AGATG-A	0.50	0.24	1.00
rs115854006	3-48346680-T-C	0.08	0.39	1.00

At the individual variant level, we observed differing effect (Figure 1). For example, for rs11082396[18-44500755-C-T] the major allele (T) was associated with reduced mosaic Y loss ($\beta_{\text{mLoY}} = 0.003$) in Wright et al. but is associated with shorter TL for both qPCR and TelSeq. Altogether, these results demonstrate that mLoY is not a major source of bias in either approach to estimating TL and that mLoY-associated variants can have different effects on TL.

Supplementary Note 4, Figure 1 Forest plot of significant variants associated with mLoY (coral) on log(OR) scale from Wright et al. with matching effect sizes from qPCR (green) and TelSeq (blue) SD scale. Variants are labelled chr-position-a0-a1 where a1 is the effect allele.

In a second analysis, we used the mosaic chromosomal alteration (mCA) data returned to UK Biobank (Field ID 3094) from Loh et al. (Loh, Genovese, and McCarroll 2020) and followed the method described in Kessler et al. (Kessler et

al. 2022) to create a binary indicator as to whether an UKB participant showed evidence of mosaic loss of X or Y. We used this to investigate whether there was an association between presence or absence of mLoX or mLoY and telomere length using a similar regression framework as described in the main manuscript to investigate clonal haematopoiesis (CH). Briefly, we removed individuals with known haem malignancies or high lymphocyte counts (>5) as well as those individuals with a somatic CH driver variant. We then used a linear model to assess whether there was an association between either mLoY or mLoX in males and females, respectively, and either qPCR or TelSeq TL metrics adjusting for age, smoking status, and ancestry PCs. We did not include sex, or related covariates given the sex specific nature of the phenotypes.

Supplementary Note 4, Figure 2 Association between mosaic chromosomal alterations and TL measurements. TL measurements from qPCR and WGS TelSeq are shown in different colours.

Overall, we observed that all classes of mosaic chromosomal alteration (mCA), were associated with shorter TL. The effect sizes between metrics exhibited heterogeneity. For example, for overall mCA, the qPCR TL point effect size estimate was less/shorter than TelSeq, concordant with expectations of

systematic differences between the metrics. However, for mLoX this was inverted, and TelSeq effect size point estimates were less/shorter.

Of relevance we note that ¹⁹ found no evidence for significant genetic correlation between qPCR LTL and mLOX/Y, which could be due to discordant effect sizes resulting in no net correlation. These results indicate a complex relationship between mLoX/mLoY and TL. Further studies will be required to fully understand the technical and biological mechanisms that underpin these observations.”

We summarise our conclusion in the main text (page 26 line 13):

“We also examined how mosaic loss of X or Y might differentially effect TelSeq and qPCR TL estimates but did not find evidence for systematic differences between the two metrics (**Supplementary Note 4**).”

1. Ding, Z. *et al.* Estimating telomere length from whole genome sequence data. *Nucleic Acids Res.* **42**, e75 (2014).
2. Taub, M. A. *et al.* Genetic determinants of telomere length from 109,122 ancestrally diverse whole-genome sequences in TOPMed. *Cell Genomics* **2**, 100084 (2022).
3. Pedersen, B. S. & Quinlan, A. R. Mosdepth: quick coverage calculation for genomes and exomes. *Bioinformatics* **34**, 867–868 (2018).
4. Erichson, N. B., Voronin, S., Brunton, S. L. & Kutz, J. N. Randomized matrix decompositions using R. *J. Stat. Softw.* **89**, (2019).
5. Codd, V. *et al.* Measurement and initial characterization of leukocyte telomere length in 474,074 participants in UK Biobank. *Nature Aging* **2**, 170–179 (2022).
6. Wang, Q. *et al.* Rare variant contribution to human disease in 281,104 UK Biobank exomes. *Nature* **597**, 527–532 (2021).

7. Sun, B. B. *et al.* Plasma proteomic associations with genetics and health in the UK Biobank. *Nature* 1–10 (2023).
8. Giambartolomei, C. *et al.* Bayesian test for colocalisation between pairs of genetic association studies using summary statistics. *PLoS Genet.* **10**, e1004383 (2014).
9. Spegg, V. *et al.* Phase separation properties of RPA combine high-affinity ssDNA binding with dynamic condensate functions at telomeres. *Nat. Struct. Mol. Biol.* **30**, 451–462 (2023).
10. Dhindsa, R. S. *et al.* Rare variant associations with plasma protein levels in the UK Biobank. *Nature* 1–9 (2023).
11. Njajou, O. T. *et al.* Telomere length is paternally inherited and is associated with parental lifespan. *Proc. Natl. Acad. Sci. U. S. A.* **104**, 12135–12139 (2007).
12. Broer, L. *et al.* Meta-analysis of telomere length in 19,713 subjects reveals high heritability, stronger maternal inheritance and a paternal age effect. *Eur. J. Hum. Genet.* **21**, 1163–1168 (2013).
13. Bountziouka, V. *et al.* Modifiable traits, healthy behaviours, and leukocyte telomere length: a population-based study in UK Biobank. *Lancet Healthy Longev.* **3**, e321–e331 (2022).
14. Ge, T., Chen, C.-Y., Ni, Y., Feng, Y.-C. A. & Smoller, J. W. Polygenic prediction via Bayesian regression and continuous shrinkage priors. *Nat. Commun.* **10**, 1776 (2019).
15. Chen, C.-Y., Han, Jgi

- ., Hunter, D. J., Kraft, P. & Price, A. L. Explicit modeling of ancestry improves polygenic risk scores and BLUP prediction. *Genet. Epidemiol.* **39**, 427–438 (2015).
16. Backman, J. D. *et al.* Exome sequencing and analysis of 454,787 UK Biobank participants. *Nature* **599**, 628–634 (2021).
17. Yu, G. & He, Q.-Y. ReactomePA: an R/Bioconductor package for reactome pathway analysis and visualization. *Mol. Biosyst.* **12**, 477–479 (2016).
18. Wright, D. J. *et al.* Genetic variants associated with mosaic Y chromosome loss highlight cell cycle genes and overlap with cancer susceptibility. *Nat. Genet.* **49**, 674–679 (2017).
19. Kessler, M. D. *et al.* Common and rare variant associations with clonal haematopoiesis phenotypes. *Nature* **612**, 301–309 (2022).c

Decision Letter, first revision:

13th May 2024

Dear Slave,

Thank you for submitting your revised manuscript "Genetic architecture of telomere length in 462,666 UK Biobank whole-genome sequences" (NG-A63425R). It has now been seen by the original referees and their comments are below. The reviewers find that the paper has improved in revision, and therefore we'll be happy in principle to publish it in Nature Genetics, pending minor revisions to satisfy the referees' final requests and to comply with our editorial and formatting guidelines.

Thank you again for your interest in Nature Genetics. Please do not hesitate to contact me if you have

any questions.

Sincerely,

Michael Fletcher, PhD
Senior Editor, Nature Genetics
ORCID: 0000-0003-1589-7087

Reviewer #1 (Remarks to the Author):

The authors have addressed the concerns raised in the prior round of review. The paper is substantially stronger as a result of the changes made.

Reviewer #2 (Remarks to the Author):

The revised manuscript by Burren, Dhindsa, Deevi et al. addresses several of the earlier concerns raised and the newly added data especially with respect to the exclusion of individuals with hematologic malignancy raised by another reviewer are a strength. Other concerns that were raised in my prior review are also addressed in the rebuttal and the revised manuscript.

I am not sure how much the proteomic analyses add especially since telomere length thresholds are necessary for acute cellular responses that may provoke DNA damage or detectable proteomic changes but this is minor in this context.

I think the acknowledgement of some of the caveats to the discussion is also a strength.

Reviewer #3 (Remarks to the Author):

The authors have addressed reviewer concerns and provide a more comprehensive analysis detailing how their PC-based measures of LTL correspond to qPCR-based and TelSeq-based measures. They have gone to extensive lengths to address my concerns regarding the influence of somatic loss-of-Y (and X), and while there remains some interesting technical and biological questions here, I do not believe it undermines the validity of their results. Additional efforts to expand supplemental tables will prove useful in enabling other investigations that leverage these results for PRS and MR approaches across traits/pathologies. Overall I believe this work represents an important advance to the field, takes a novel approach to jointly capturing multiple aspects of LTL, identifies new common and rare variant associations, and will be of high interest to the readership.

Author Rebuttal, first revision:

Reviewer #1 (Remarks to the Author):

The authors have addressed the concerns raised in the prior round of review. The paper is substantially stronger as a result of the changes made.

We thank the reviewer for their comments and suggestions and agree that these changes have strengthened the paper substantially.

Reviewer #2 (Remarks to the Author):

The revised manuscript by Burren, Dhindsa, Deevi et al. addresses several of the earlier concerns raised and the newly added data especially with respect to the exclusion of individuals with hematologic malignancy raised by another reviewer are a strength.

Other concerns that were raised in my prior review are also addressed in the rebuttal and the revised manuscript.

I am not sure how much the proteomic analyses add especially since telomere length thresholds are necessary for acute cellular responses that may provoke DNA damage or detectable proteomic changes but this is minor in this context.

We hope that our analyses to link variants to genes and proteins in a causal framework will prove useful for researchers interested in potential biological mechanisms underlying the modulation of telomere length and potential clinical lines of sight. For example, a missense variant p.V762A - rs1136410 in PARP1 has been linked to protection from some subtypes of CH [PMC10438798], and through our analyses we have been able to causally link lower PARP1 plasma protein abundance to lower telomere length, which may strengthen the rationale for the use of PARP inhibitors in reversing specific CH subtypes.

I think the acknowledgement of some of the caveats to the discussion is also a strength.

We thank the reviewer for their comments and suggestions.

Reviewer #3 (Remarks to the Author):

The authors have addressed reviewer concerns and provide a more comprehensive analysis detailing how their PC-based measures of LTL correspond to qPCR-based and TelSeq-based measures. They have gone to extensive lengths to address my concerns regarding the influence of somatic loss-of-Y (and X), and while there remains some interesting technical and biological questions here, I do not believe it undermines the validity of their results. Additional efforts to expand supplemental tables will prove useful in enabling other investigations that leverage these results for PRS and MR approaches across traits/pathologies. Overall I believe this work represents an important advance to the field, takes a novel approach to jointly capturing multiple aspects of LTL, identifies new common and rare variant associations, and will be of high interest to the readership.

We thank the reviewer for their comments and suggestions and hope that the expansion of the supplementary tables alongside our investigations into LoX and LoY will prove enabling to other researchers.

Final Decision Letter:

25th Jul 2024

Dear Slave,

I am delighted to say that your manuscript "Genetic architecture of telomere length in 462,666 UK Biobank whole-genome sequences" has been accepted for publication in an upcoming issue of Nature Genetics.

Your paper will be published online after we receive your corrections and will appear in print in the

next available issue. You can find out your date of online publication by contacting the Nature Press Office (press@nature.com) after sending your e-proof corrections.

Please note that *Nature Genetics* is a Transformative Journal (TJ). Authors may publish their research with us through the traditional subscription access route or make their paper immediately open access through payment of an article-processing charge (APC). Authors will not be required to make a final decision about access to their article until it has been accepted. Find out more about Transformative Journals

Authors may need to take specific actions to achieve compliance with funder and institutional open access mandates. If your research is supported by a funder that requires immediate open access (e.g. according to Plan S principles) then you should select the gold OA route, and we will direct you to the compliant route where possible. For authors selecting the subscription publication route, the journal's standard licensing terms will need to be accepted, including <https://www.nature.com/nature-portfolio/editorial-policies/self-archiving-and-license-to-publish>. Those licensing terms will supersede any other terms that the author or any third party may assert apply to any version of the manuscript.

If you have not already done so, we strongly recommend that you upload the step-by-step protocols used in this manuscript to protocols.io. protocols.io is an open online resource that allows researchers to share their detailed experimental know-how. All uploaded protocols are made freely available and are assigned DOIs for ease of citation. Protocols can be linked to any publications in which they are used and will be linked to from your article. You can also establish a dedicated workspace to collect all your lab Protocols. By uploading your Protocols to protocols.io, you are enabling researchers to more readily reproduce or adapt the methodology you use, as well as increasing the visibility of your protocols and papers. Upload your Protocols at <https://protocols.io>. Further information can be found at <https://www.protocols.io/help/publish-articles>.

Sincerely,

Michael Fletcher, PhD
Senior Editor, Nature Genetics
ORCID: 0000-0003-1589-7087